# EFFECTIVE DATA PRUNING THROUGH SCORE EXTRAPOLATION

## ABSTRACT

Training advanced machine learning models demands massive datasets, resulting in prohibitive computational costs. To address this challenge, data pruning techniques identify and remove redundant training samples while preserving model performance. Yet, existing pruning techniques predominantly require a full initial training pass to identify removable samples, negating any positive benefit in most scenarios. To overcome this limitation, we introduce a novel importance score extrapolation framework that requires training on only a small subset of data. We present two initial approaches in this framework[1]—k-nearest neighbors and graph neural networks—to accurately predict sample importance for the entire dataset using patterns learned from this minimal subset. We demonstrate the effectiveness of our approach for 2 state-of-the-art pruning methods (Dynamic Uncertainty and TDDS), 4 different datasets (CIFAR-10, CIFAR-100, Places-365, and ImageNet), and 3 training paradigms (supervised, unsupervised, and adversarial). Our results indicate that score extrapolation is a promising direction to scale expensive score calculation methods, such as pruning, data attribution, or other tasks.

## 1 INTRODUCTION

In recent years, the demand for large and comprehensive datasets has grown rapidly. This is particularly evident in the development of advanced models, such as large language models (LLMs) (Minaee et al., 2024), and other forms of foundation models (Kirillov et al., 2023; Götz et al., 2025b;a), which require vast amounts of data to train.

In this context, dataset pruning has emerged as a valuable technique to optimize the training process and improve the efficiency of model development. By scoring individual data points by their importance and only selecting the most informative, dataset pruning aims to improve training efficiency while maintaining model performance. This approach is particularly beneficial in scenarios where the available dataset is vast, and the computational resources required for training the model on the entire dataset are significant, such as autonomous driving (Schmidt et al., 2020; 2024).

Yet, the majority of existing pruning approaches fail to deliver positive efficiency gains in practice, as they either require computationally expensive procedures (Griffin et al., 2024) or necessitate training a model on the full dataset to derive importance scores (He et al., 2024; Zhang et al., 2024), which is inconsistent with the central motivation of data pruning.

To address the inherent problem of time inefficiency in data pruning approaches, we formulate the research question: **"Can importance scores for unseen samples be efficiently extrapolated from a small subset of known scores?"**. To explore this question, we propose a framework for score extrapolation, limiting expensive score computation to a small initial subset of the full training data. We propose one training-free and one training-dependent extrapolation approach and demonstrate that extrapolation can make data pruning truly efficient across multiple tasks and scoring functions. Beyond immediate pruning applications, our findings suggest that importance score extrapolation offers a scalable approach that may be applied to other techniques involving expensive per-sample evaluations, including data attribution methods (Koh & Liang, 2017; Ilyas et al., 2022).

---

[1]`https://anonymous.4open.science/r/extra-3138`

Our contribution can be summarized as follows:

- We examine the feasibility of the suggested *score extrapolation* paradigm and introduce a framework to extrapolate expensive importance scores to unseen data samples, *significantly reducing computational effort*.
- In the scope of this framework, we introduce two extrapolation techniques based on k-nearest-neighbors (KNN) and graph neural networks (GNN).
- In an extensive empirical study, we showcase the effectiveness of our extrapolation-based approaches for **2** state-of-the-art pruning methods, i.e., Dynamic Uncertainty (DU) (He et al., 2024), Temporal Dual-Depth Scoring (TDDS) (Zhang et al., 2024), **4** datasets (CIFAR-10, CIFAR-100 (Krizhevsky et al., 2009), Places-365 (Zhou et al., 2017), ImageNet (Deng et al., 2009)), and **3** different training paradigms (supervised, unsupervised, adversarial).

## 2 RELATED WORK

**Data Pruning.** Data pruning, or coreset selection (Sorscher et al., 2022; He et al., 2024; Zhang et al., 2024; Guo et al., 2022), aims to keep a small, representative subset of training data that preserves model performance while reducing computational costs. Several efficient strategies have emerged, which can be grouped into three categories (Tan et al., 2023) and either estimate importance or difficulty scores, use geometric calculation, or employ optimization. Importantly, *approaches usually require full training on the dataset* (Zhang et al., 2024; He et al., 2024; Toneva et al., 2019) to estimate importance scores or latent space representations.

**Pruning Based on Importance.** Methods from this category assign scores to samples based on their utility for training, typically retaining the highest-ranked examples. Techniques include GradNd (Paul et al., 2021) or TDDS (Zhang et al., 2024), which utilize gradients, while others, like EL2N (Paul et al., 2021), rely on prediction errors. Coleman et al. (2020) use entropy from proxy models for ranking, and forgetting scores track incorrect classifications, indicating that frequently forgotten samples are informative. DU (He et al., 2024) assesses uncertainty through the standard deviation of predictions during training. Alternatively, importance can be measured by the impact of sample removal. MoSo (Tan et al., 2023) calculates the change in empirical risk, while (Feldman & Zhang, 2020; Yang et al., 2023) use influence functions (Koh & Liang, 2017) to gauge the influence of a sample on generalization performance. *This category is usually the most effective but requires costly training on the full dataset to estimate scores*.

**Pruning Based on Geometry.** Other approaches leverage geometric or data distribution properties. Herding (Welling & Bren, 2009; Chen et al., 2010) and Moderate (Xia et al., 2022) compute distances in feature space. Others (Huang et al., 2024; Mirzasoleiman et al., 2020) combine gradients with distances. Har-Peled et al. (2006) forms a coreset by approximating the maximum margin hyperplane. Self-Supervised Pruning (SSP) (Sorscher et al., 2022) uses $k$-means clustering in self-supervised model embeddings, ranking samples by cosine similarity to centroids. $k$-center methods (Sener & Savarese, 2018; Griffin et al., 2024) minimize the maximum distance from samples to centers, while coverage-based methods (Zheng et al., 2023) enhance sample diversity and reduce redundancy.

**Pruning Based on Optimization.** Borsos et al. (2020) used greedy forward selection to solve a cardinality-constrained bilevel optimization problem for subset selection. GLISTER (Killamsetty et al., 2021c) performs coreset selection via a greedy-Taylor approximation of a bilevel objective while simultaneously updating model parameters. GradMatch (Killamsetty et al., 2021a) defines and minimizes a gradient error using the Orthogonal Matching Pursuit algorithm.

**Adverserial Robustness.** Data pruning in adversarial training remains relatively underexplored. Existing methods can be grouped into three broad categories. The first focuses on dynamically selecting which samples to apply adversarial perturbations to during training (Hua et al., 2021; Chen & Lee, 2024). The second includes coreset-based methods to reduce training data while preserving robustness (Killamsetty et al., 2021b; Dolatabadi et al., 2023). The third relies on heuristic pruning strategies to discard samples less useful for robust learning (Kaufmann et al., 2022; Li et al., 2023).

**Orthogonal Works.** Beyond data pruning, identifying important data subsets is a fundamental problem in various machine learning paradigms, including continual learning (Wang et al., 2024), data distillation (Wang et al., 2018; Holder & Shafique, 2021), and active learning (Kirsch & Gal,

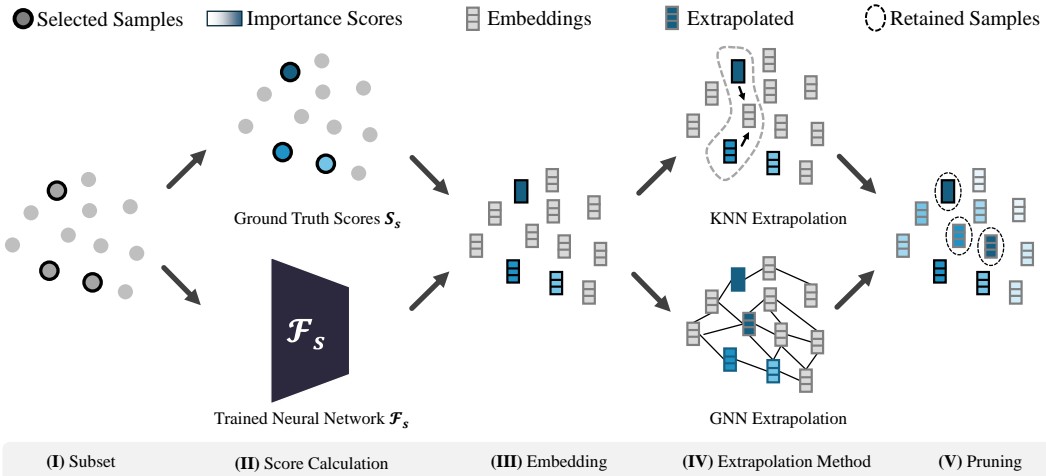

Figure 1: Extrapolation concept overview. **(I)** We start by randomly selecting a subset $\mathbb{D}_s$ of $m$ samples out of the full dataset $\mathbb{D}$ compromising $n$ samples (where $m << n$). **(II)** We train a neural network $F_s$ on the selected subset and calculate ground truth importance scores $S_S$ with the selected pruning method during the training run. **(III)** Using the trained model, we map the whole dataset $\mathbb{D}$ to the embedding space of the network. **(IV)** Subsequently, we select an extrapolation method and train it to extrapolate scores on $\mathbb{D}_s$ and the calculated ground truth scores $S_s$. **(V)** Finally, we use the original and extrapolated scores to perform the pruning task, selecting the top-k samples exhibiting the highest importance scores.

2022). While active learning focuses on selecting samples to label, data pruning addresses the challenge of selecting which samples to retain for training.

Data pruning methods lead to effective data reduction. However, approaches usually require training on the full dataset or complex optimization to estimate which samples to prune. Both can exceed full model training time. Only a few works address efficiency. RS2 (Okanovic et al., 2024) proposes repeated random sampling during training. Similarly, works like InfoBatch Qin et al. (2024) instance-dependent early stopping Yuan et al. (2025) continued the idea and dropped samples earlier during the training, which can be referred to as *dynamic pruning*. Tang et al. (2023) selects data online based on gradient evaluations, while other approaches rely on proxy models (Coleman et al., 2020). *The prohibitively expensive training on the full datasets is an unaddressed problem, which we address in this work through score extrapolation.*

## 3 SCORE EXTRAPOLATION

Existing data pruning methods (Zhang et al., 2024; He et al., 2024; Paul et al., 2021) usually require complete training on the full dataset or rely on costly optimizations (Killamsetty et al., 2021c), which makes them prohibitively expensive. As we show in Section 4, even zero-shot methods relying on foundation models and distance calculation are time-inefficient.

To address the computational challenges of estimating data importance scores for large-scale datasets, we propose a score extrapolation framework. Within this framework, we compute the importance scores *only* on a minor subset of the large-scale dataset and extrapolate scores for the remaining data points efficiently. The concept is shown in Figure 1 and comprises the following steps: At first, a subset of the large-scale dataset is selected **(I)**. Based on this subset, a model is trained to calculate importance scores using a chosen scoring method **(II)**. To extrapolate the scores of the remaining samples, we generate the embeddings of the dataset based on the trained model **(III)**. With the calculated scores of the subset and the embeddings, we use one of the proposed extrapolation methods based on either KNN or a GNN **(IV)** to estimate scores for the remaining samples. In the last step **(V)**, we perform the regular pruning task with our extrapolated scores.

**Subset Definition.** Instead of calculating the importance score for each sample $x$ in the dataset, *which requires a full training*, we aim to select a representative training subset $\mathbb{D}_s \subset \mathbb{D}$ of cardinality $m$ from the full dataset $\mathbb{D}$, comprising $n$ samples, where $m << n$. For the residual set $\mathbb{D}_r = \mathbb{D} \setminus \mathbb{D}_s$,

we will extrapolate the scores later. An initial selection of $\mathbb{D}_s$ based on a coverage radius or density neighborhood requires intensive training or calculation and might not be consistently effective. Thus, we propose a uniform iid sampling, which provides an unbiased estimate of the underlying data distribution and converges to the true distribution with an increased number of samples (Robert & Casella, 2004). We further examine the initial selection strategy in Appendix C.1.

$$\mathbb{D}_s = \{x_1, x_2, \ldots, x_m\}, \quad \text{where} \quad x_i \overset{\text{i.i.d.}}{\sim} \text{Uniform}(\mathbb{D}), \quad \forall i \in \{1, \ldots, m\}$$

**Model Training and Score Calculation.** For our extrapolation, we need to choose a pruning method $\Lambda$ to calculate the associated importance scores $S$ for data pruning. *Note that the quality of extrapolated scores will be inherently upper-bounded by the chosen method.* We then train a model $\mathcal{F}_s$ on the defined training subset $\mathbb{D}_s$, using the same setup as the pruning method would normally use for the full dataset. Through the training process, we obtain ground truth data importance scores $S_s = \Lambda_s(x)_{x \in \mathbb{D}_s} \in \mathbb{R}^m$ for the subset $\mathbb{D}_s$. After training, the encoder $\phi_s(\cdot)$ of $F_s$ maps all samples in $\mathbb{D}$ to embeddings $z = \phi_s(x) \in \mathbb{R}^d$, which serve as input features for extrapolation.

**Extrapolation through Interpolation.** Based on the initial computed scores $S_s$, we apply an extrapolation schema to generate the scores $S_r \in \mathbb{R}^{n-m}$ for the residual set $\mathbb{D}_r$. In this work, we present two approaches to extrapolate importance scores based on the embedding information of a sample $x$, which we motivate through the influence function literature (Koh & Liang, 2017).

*Theoretical Motivation.* Influence functions are a classical tool from robust statistics that measure how a model's parameters and, therefore, its predictions change under changes in the training data (Koh & Liang, 2017). To compute sample importance, influence functions require only access to model gradients and Hessian-vector products. In the following, we establish a connection between the influence function literature and the properties of our scoring approaches. For a neural network $\mathcal{F}$ with embedding function $\phi : \mathcal{X} \to \mathbb{R}^d$ that maps inputs $x \in \mathcal{X}$ to embeddings $z = \phi(x)$, we assume linearity locally around samples, e.g., reference point $x_0$ (Szegedy et al., 2013):

$$\phi_\theta(x) \approx \phi_\theta(x_0) + J_\phi(x_0)(x - x_0), \tag{1}$$

where $J_\phi(x_0)$ is the Jacobian of the embedding with respect to the input.
The influence function $\mathcal{I}(\cdot, \cdot)$ of the last layer acting as a linear classifier on the embeddings parameterized by $\theta$ and labels $y \in \{-1, 1\}$ can be given by (Koh & Liang, 2017):

$$\mathcal{I}(z, z_{\text{test}}) = -y_{\text{test}} y \, \sigma\big(-y_{\text{test}} \theta^\top z_{\text{test}}\big) \, \sigma\big(-y \theta^\top z\big) \, \big(z_{\text{test}}^\top H_\theta^{-1} z\big), \tag{2}$$

where $\sigma(\cdot)$ is the sigmoid function, $H_\theta$ is the empirical Hessian at the optimum, $z$ is the embedding of a training point $x$, and $z_{\text{test}}$ the embedding of an evaluation point (e.g., a sample that we want to assess the importance of). Here, the importance is linearly dependent on the embedding $z_{test}$ of the sample $x_{test}$ except for a smooth scalar multiplier:

$$s(z, z_{\text{test}}) := -y_{\text{test}} y \, \sigma\big(-y_{\text{test}} \theta^\top x_{\text{test}}\big) \, \sigma\big(-y \theta^\top x\big). \tag{3}$$

If $x_1, x_2$ are close and linearization holds, then for a convex interpolation $x_\lambda = \lambda x_1 + (1 - \lambda)x_2$,

$$\phi_\theta(x_\lambda) \approx \lambda\phi_\theta(x_1) + (1 - \lambda)\phi_\theta(x_2), \tag{4}$$

$$\mathcal{I}(z, z_\lambda) \approx \lambda\mathcal{I}(z, z_1) + (1 - \lambda)\mathcal{I}(z, z_2), \tag{5}$$

up to smooth variations in $s(z, z_{\text{test}})$. This result motivates why the influence of a sample may interpolate smoothly in the embedding space for nearby points.

*Practical Considerations for Extrapolation.* The equations above state that for two nearby evaluation points, the influence on their convex combination is approximately a convex combination of their respective influences (under local linearity assumptions). This does not assert that influence is a radial function of the Euclidean distance to reference points. However, our extrapolation methods adopt this smoothness, and nearby samples tend to share similar importance as a practical prior. We support our theoretical motivation with empirical examination in Sec. 4 and App. C.5.

**Extrapolation with KNN.** To extrapolate scores, we propose a simple KNN-based approach that functions as a baseline for more complex methods. We start from the subset $\mathbb{D}_s$ and its associated importance scores $S_s$. Next, we utilize the encoder $\phi_s(x) : \mathcal{X} \to \mathbb{R}^d$ of our trained model $\mathcal{F}_s$ to transform the input samples $x \in \mathcal{X}$ into the $d$-dimensional embedding space of the encoder. Then, we compute the score for each data point $x \in \mathbb{D}_r$ as the average of the scores of its $k$ nearest neighbors

in the embedding space. Specifically, the extrapolated importance scores $S_{\text{knn}}$ can be estimated with our approximation $\Lambda_{\text{knn}}(x) : \mathbb{D}_r \to S_{\text{knn},r} \in \mathbb{R}^{n-m}$ for a sample $x$ as:

$$\Lambda_{knn}(x) = \frac{\sum_{i=1}^{k} \exp(-D(\phi_s(x), \phi_s(x_{\pi_i(x)}))) S_s[\pi_i(x)]}{\sum_{i=1}^{k} \exp(-D(\phi_s(x), \phi_s(x_{\pi_i(x)})))}; \qquad \pi_i(x) \in [m]; \qquad x \in \mathbb{D}_r, \quad (6)$$

where $\pi_i(x)$ represents the index of the $i$-th nearest neighbor of $x$, and $D(\cdot, \cdot)$ denotes a chosen distance metric (i.e., Euclidean distance). This weighted average, based on the structure of the embedding space, ensures that the extrapolated scores maintain the local structure of the dataset.

**Extrapolation with GNN.** While KNN-based extrapolation serves as a simple approach to extrapolate information from the sampled subset $\mathbb{D}_s$ to the residual data $\mathbb{D}_r$, it lacks the ability to model complex interactions among data points. To address this limitation, we additionally propose a more powerful extrapolation method based on GNNs that can capture higher-order relationships in the dataset through message passing. We construct an undirected graph $\mathcal{G} = (\mathcal{V}, \mathcal{E})$, where each sample in $\mathbb{D}$ represents a vertex in $\mathcal{V}$, and edges $\mathcal{E}$ are formed between each sample and its $k$ nearest neighbors in the embedding space. The node features $h_i$ are defined on the sample embeddings $h_i = z_i = \phi_s(x_i)$. Optionally, the features can be concatenated with the one-hot encoded class labels $l$ for supervised tasks $h_i = [\phi_s(x_i), l_i]$. A study on the impact of labels on the embeddings is given in App. B.4. We define an adjacency matrix $\mathcal{A}$, comprising of edge weights $w_{ij}$ based on the latent space euclidean distance $D$ to its neighbors encoded as $w_{ij} = \exp(-D(\phi_s(x_i), \phi_s(x_{\pi_j(x_i)}))); j \in [0...k]$ for the $k$ nearest neighbors and add a self loop $w_{ii} = 1$. $\mathcal{A}$ is normlized acording to Kipf & Welling (2017).

To learn the interactions between the data samples, we employ a simple GNN $\mathcal{F}_{\mathcal{G}}(\mathcal{A}, \mathcal{V}; \theta)$ with weights $\theta$ that consists of three layers of Graph Convolutional Networks (GCNs) as described by (Kipf & Welling, 2017) and directly predicts the importance score of the sample nodes given our defined node features $H = [h_1...h_n] \in \mathbb{R}^{n \times d}$ of the vertices $\mathcal{V}$ and adjacency matrix $\mathcal{A}$. To scale training in large graphs, we employ neighbor sampling (Hamilton et al., 2017) to generate mini-batches of nodes and their local neighborhoods. Based on these input mini-batches of the graph, the GNN outputs a vector of predicted scores $\mathcal{F}_{\mathcal{G}}(\mathcal{A}, \mathcal{V}; \theta) \to S_{\text{GNN}} \in \mathbb{R}^n$, reflecting a scalar importance score for each node. We refer to the importance score of node $i$ with $\mathcal{F}_{\mathcal{G}}(\mathcal{A}, \mathcal{V}; \theta)_i$. Since we only have the reference score $S_s$ for samples in the training dataset $\mathbb{D}_s$, we compute the mean square error loss, selectively only over the nodes reflecting samples in $\mathbb{D}_s$:

$$\mathcal{L} = \frac{1}{|\mathbb{D}_s|} \sum_{i \in [m]} \left( \mathcal{F}_{\mathcal{G}}(\mathcal{A}, \mathcal{V}; \theta)_i - S_s[i] \right)^2, \qquad (7)$$

where $\mathcal{F}_{\mathcal{G}}(\mathcal{A}, \mathcal{V}; \theta)_i$ is the prediction for index $i$. After training, we use the GNN to infer scores for all samples in $\mathbb{D} \setminus \mathbb{D}_s$, such that $S_{\text{GNN},r} = [\mathcal{F}_{\mathcal{G}}(\mathcal{A}, \mathcal{V}; \theta)_j]; j \in \{[n] \setminus [m]\}$. The message passing in the GNN allows the model to leverage the structural information of the entire dataset, potentially leading to more accurate extrapolation of the scores. Importantly, GNNs are *significantly less computationally expensive than the task model*, which we analyze in our experiments.

Based on the proposed extrapolation schemes, we assigned importance scores by extrapolating the scores for $\mathbb{D}_r$ and retaining the directly computed ground truth values for $\mathbb{D}_s$. Then, the final score vector $S = [S_s, S_r]$ is concatenated from the original scores $S_s$ and the extrapolated scores $S_r$ consisting of either our KNN scores $S_r = S_{\text{knn},r}$ or for the GNN $S_r = S_{\text{GNN},r}$. Once scores are computed for the full dataset, they can be used in the respective downstream tasks. In this work, we specifically focus on data pruning. However, the general framework can also be applied to *other settings requiring sample-level importance scores*, such as *data attribution* (Koh & Liang, 2017; Ilyas et al., 2022). This strategy significantly reduces the computational cost of scoring large datasets. Instead of training on the full dataset, scores are computed only on a small subset, while the remaining samples are efficiently approximated through extrapolation.

## 4 EVALUATION

In the following section, we evaluate our importance score extrapolation paradigm for different pruning methods, tasks, and datasets. The primary objective of this paper is to evaluate the research question *"Can importance scores for unseen samples be efficiently extrapolated from a small subset of known scores?"* and to investigate the strengths and limitations of this approach. Our experiments

are designed to analyze the core properties of the method rather than to maximize performance metrics. We have three main goals for score extrapolation: **1)** *reduce* the computation time compared to standard pruning, **2)** *maintain* downstream task performance, and **3)** show high *correlation* to the original scores which are generated by training a model and estimating the scores on the full set $\mathbb{D}$. Since the research question aims to address the practical problem of importance scores estimation, which always requires full training, we focus on efficiency. To examine these properties, we evaluate the tasks of classic data pruning for supervised-, unsupervised-, and adversarial training.

**Scores.** In our experiments, we extrapolate two state-of-the-art pruning methods, DU (He et al., 2024), TDDS (Zhang et al., 2024) and DUAL (Cho et al., 2025). These methods require training on the full dataset for several epochs to obtain reliable scores. They also store logits per sample at each epoch; DU retains only the softmax logits for the correct class, whereas TDDS approximates gradients from full logit outputs, making it more memory-intensive. We focus on the importance score-based method, as hybrid or geometric methods like CCS (Zheng et al., 2023) require expensive distance calculations, counteracting the speedup of extrapolation, further details on scores in App. B.

**Supervised Data Pruning.** We first evaluate score extrapolation on supervised data pruning. The objective of this task is to minimize the amount of training data while maintaining model performance as much as possible. Since we focus on large datasets, we use synthetic CIFAR-100 1M (Krizhevsky et al., 2009; Wang et al., 2023b), Places 365 (Zhou et al., 2017), and ImageNet 1k (Deng et al., 2009). We compare our extrapolated scores with the original pruning approaches, random pruning, and the training-free approaches ZCoreSet (Griffin et al., 2024) and SSP (Sorscher et al., 2022), and examine the different initial subset sizes $\mathbb{D}_s$ ranging from 10% to 20%. If not stated otherwise, we use 20% of the full dataset for $\mathbb{D}_s$ in all experiments. More details are given in Appendix A. All experiments were conducted with 3 different random initializations.

Table 1 illustrates the effect on relative accuracy (since extrapolation approximates the original scores) and time efficiency of our extrapolation schemes across different datasets and pruning approaches. The time measurement includes *all* steps, for regular pruning as well as our extrapolation, including the initial training for score estimation, possible scores extrapolations, and a training on the pruned subset (see Appendix A.6). The results show that extrapolated scores maintain the high quality of the original scores *close to their maximum pruning rate*. However, score extrapolation is considerably faster, *as it does not require training on all data samples for the score estimation*.
Notably, the 20% training subsets $\mathbb{D}_s$ demonstrate strong performance, and the smaller 10% subset also shows decent results and generally outperforms random pruning. GNN-based extrapolation generally performs better than using KNN. However, KNN extrapolation is *Pareto optimal* w.r.t. time-accuracy trade-offs (see Time Optimality) and reduces computation costs by up to 4.9x (see *).

**Pruning Performance.** In Figure 2 we compare pruning strategies at different pruning rates. The difference in final accuracy between ground truth scores to extrapolation methods is low for Places365 (Figure 2a) and ImageNet (Figure 2b). While TDDS maintains a higher accuracy for moderate pruning rates, the difference to our extrapolation decreases for higher pruning rates. DU performs worse than random pruning for pruning rates over 50%, making comparisons in this regime irrelevant.
For synthetic CIFAR-100 1M, the original DU and TDDS actually improve model accuracy compared to standard training. We attribute this to the ability of the pruning methods to filter out noisy data.

Table 1: Relative accuracy (%) to the original accuracy (100%), *at the maximum pruning rate where the original score still outperforms random pruning*; improvements over Random pruning in percent points (pp); inference time in minutes (± std. error) in gray, best score approximation in bold.

| Dataset | Prune % | Method | Original | GNN 20% | KNN 20% | GNN 10% | KNN* 10% | Random |
|---|---|---|---|---|---|---|---|---|
| ImageNet | 50 | DU | 100% 1367 ± 19 | **99.68% (+0.56pp)** 829 ± 15 | 99.43% (+0.31pp) 687 ± 16 | 98.68% (-0.44pp) 701 ± 12 | 99.04% (-0.08pp) **581 ± 10** | 99.12% – |
| Places365 | 50 | DU | 100% 2416 ± 26 | **99.21% (+0.87pp)** 1631 ± 18 | 98.71% (+0.37pp) 1330 ± 14 | 98.76% (+0.42pp) 1394 ± 15 | 98.86% (+0.52pp) **1075 ± 13** | 98.34% – |
| Places365 | 95 | TDDS | 100% 1741 ± 22 | **98.12% (+3.08pp)** 917 ± 17 | 96.96% (+1.92pp) 621 ± 13 | 97.02% (+1.98pp) 668 ± 11 | 96.19% (+1.15pp) **353 ± 8*** | 95.04% – |
| Synthetic CIFAR-100 | 95 | DU | 100% 412 ± 9 | **97.48% (+3.15pp)** 374 ± 6 | 96.79% (+2.46pp) 158 ± 5 | 96.05% (+1.72pp) 333 ± 6 | 96.34% (+1.01pp) **114 ± 5** | 94.33% – |
| Synthetic CIFAR-100 | 95 | TDDS | 100% 441 ± 10 | **98.13% (+5.22pp)** 387 ± 6 | 97.64% (+4.73pp) 168 ± 6 | 95.49% (+2.58pp) 343 ± 7 | 95.27% (+2.36pp) **122 ± 5** | 92.91% – |

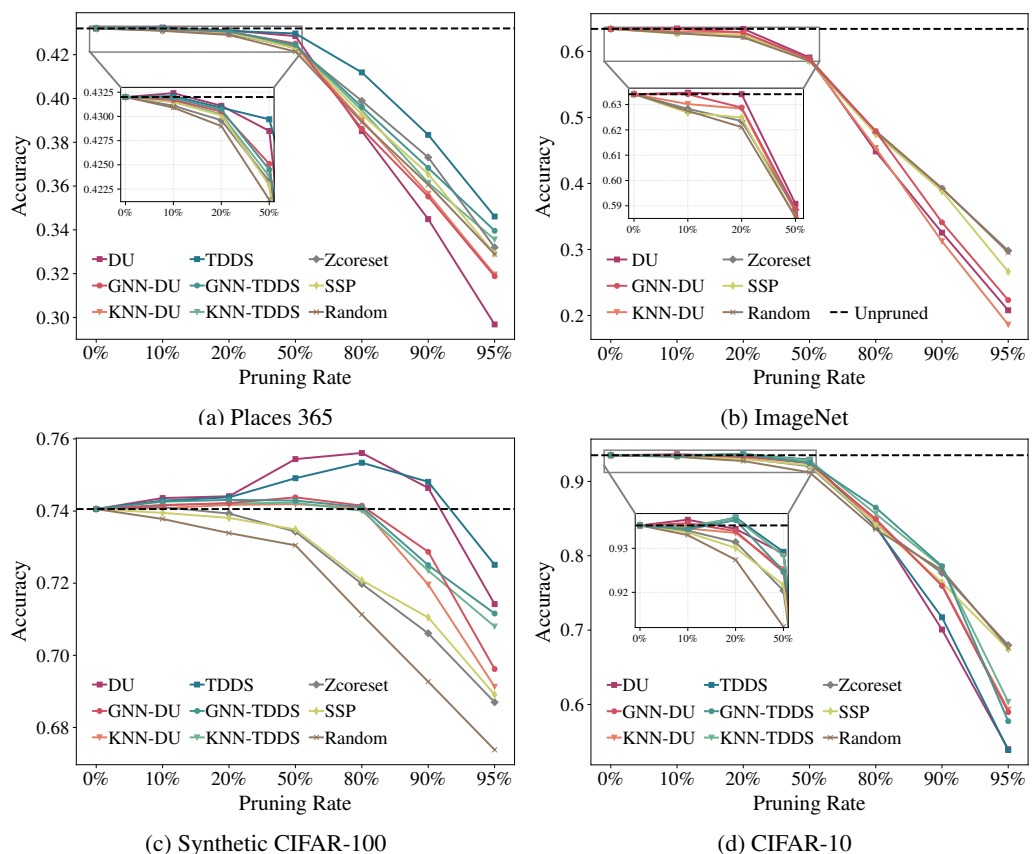

Figure 2: Downstream accuracy as a function of pruning rate for multiple datasets, averaged over three seeds. GNN-based extrapolation generally yields the strongest results, with marginal differences from standard scores at moderate pruning rates. For each dataset, importance scores are computed on a 20% subset (40% for CIFAR-10), followed by extrapolation. Extrapolated pruning surpasses Zcoreset, SSP, and Random when the original scores do.

Similarly, score extrapolation techniques demonstrate performance gains over standard training, though the effect is less substantial than with direct pruning methods. Especially for this dataset, our extrapolated scores strongly outperform the training-free approaches.

For the smaller CIFAR-10 (Figure 2d), the performance between ground truth scores and extrapolated ones is identical for TDDS up to 20% pruning. Importantly, across all datasets, *extrapolated scores consistently outperform random and training-free pruning whenever the ground truth score does*, demonstrating the practical value of score extrapolation.

**Time Optimality.** We examine the computational behavior of the extrapolation approaches in more detail. Therefore, we track in a Pareto plot *all steps* (from score calculation to the final training) required to train on the pruned dataset. All steps for extrapolation include: *initial training* and *score estimation on the subset, embedding calculation* and *extrapolation*, and *final evaluation training on the pruned set*. For the regular methods, we track the *training* required for the score calculation, the *score calculation* itself, and again, the *final evaluation training*. Further details and an extended analysis are provided in Appendix A.6. While the original scores demonstrate higher accuracy and can be viewed as an upper bound for the extrapolation methods, they require a full training for score estimation, eliminating any practical benefits in most cases. Figure 3 compares the model performance, including the pruning, extrapolation, and training time. We see that our extrapolation methods, especially the KNN extrapolation, are always Pareto optimal for Places365 and ImageNet, and even faster than the training-free approaches ZCoreSet and SSP. Our extrapolated scores provide a time advantage already for a single model training, while the standard pruning approaches only obtain any real efficiency advantage if multiple models are trained with the pruned data.

**Relationship between Score Correlation and Downstream Task Performance.** To verify the results presented in Table 1, we examine the correlation between the extrapolation and the original

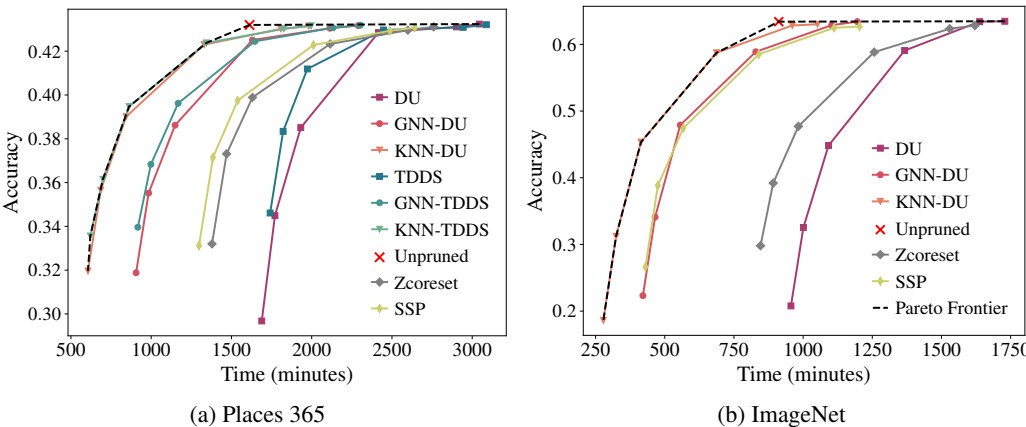

(a) Places 365        (b) ImageNet

Figure 3: Computation time versus final test accuracy for various pruning strategies, including training without pruning. Our extrapolation methods, particularly KNN, consistently achieve optimal trade-offs between efficiency and performance, demonstrating the Pareto-optimality of our extrapolation approaches. Results use 20% scored subset for extrapolation. Points reflect different pruning rates.

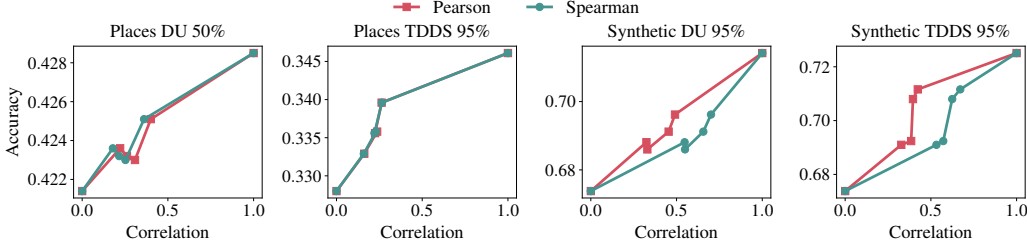

Figure 4: Analysis of the dependency of correlation and accuracy of the extrapolation methods for Places365 and synthetic CIFAR-100. The accuracy increases with the correlation of the specific task. Each point reflects a different subset size from 0 (pure random) to 100 (pure method).

scores. Table 2 reports Pearson (Benesty et al., 2009) and Spearman (Zar, 2005) correlation. As expected, the correlation increases with the subset size. In the previous pruning results, we saw that a 10% subset size is already sufficient for high performance. In addition, the GNN's correlation is always higher than the KNN's, underlining the GNN's greater ability to capture dataset properties.

Next, we investigated the actual relationship of these correlation scores to the downstream task performance. In Figure 4, we compare correlation and accuracy. The correlation of the original and extrapolated scores is perfectly aligned with the final accuracy, confirming the importance of accurately extrapolating scores.

**Unsupervised Pruning Score Extrapolation.** Here, we investigate the applicability of score extrapolation to unsupervised training settings beyond the supervised training and pruning scenarios explored in the previous experiments. When dealing with large datasets, the assumption that all data is labeled might not be realistic. Thus, we modify the original data pruning task to work with unlabeled datasets. We use Tur-

Table 2: Pearson and Spearman correlations for different pruning methods across datasets. GNN consistently outperforms KNN, and performance improves with larger subset sizes.

| Dataset | Method | Subset Size (%) | Pearson $\rho$ | | Spearman $r_s$ | |
|---|---|---|---|---|---|---|
| | | | **GNN** | **KNN** | **GNN** | **KNN** |
| Imagenet | DU | 20 | **0.4193** | 0.3779 | **0.3503** | 0.3068 |
| | | 10 | **0.2850** | 0.2575 | **0.2178** | 0.1980 |
| Places365 | DU | 20 | **0.4004** | 0.3081 | **0.3611** | 0.2524 |
| | | 10 | **0.2612** | 0.2215 | **0.2158** | 0.1791 |
| | TDDS | 20 | **0.2632** | 0.2251 | **0.2646** | 0.2214 |
| | | 10 | **0.2372** | 0.1620 | **0.2297** | 0.1594 |
| Synthetic CIFAR | DU | 20 | **0.4910** | 0.4538 | **0.7009** | 0.6562 |
| | | 10 | **0.3396** | 0.3243 | **0.5593** | 0.5471 |
| | TDDS | 20 | **0.4236** | 0.3955 | **0.6713** | 0.6244 |
| | | 10 | **0.3849** | 0.3273 | **0.5722** | 0.5324 |

tle (Gadetsky et al., 2024) for the unsupervised image classification on CIFAR-10, and apply the score calculation with DU and our extrapolations. Further details on are given in App A and B.3.

For our newly created unsupervised data pruning setup, we focus on analyzing the correlation of our extrapolation approaches to the original DU score. In Table 3, we observe that the correlations of the extrapolated scores are as high as for the standard pruning approach and increase with the size of the subset, indicating the flexibility of the extrapolation paradigm. Interestingly, KNN has a higher correlation than GNN. This can be explained by the linear assumption made in Turtle, which favors the simple linear extrapolation, compared to the, in this case, overcomplex GNN.

**Adversarial training.** In addition to the supervised and unsupervised, we investigate adversarial training as another setting for score extrapolation. In this setting, we aim to select a minimal dataset to maintain a high performance on unperturbed data (clean data accuracy), while being robust against attacks (accuracy under perturbations). We perform adversarial training in the $\ell_\infty$-norm with a perturbation budget of

Table 3: Pearson and Spearman correlations for unsupervised extrapolation for CIFAR10 with DU based on the Turtle (Gadetsky et al., 2024).

| Subset size | Pearson $\rho$ | | Spearman $r_s$ | |
|---|---|---|---|---|
| | **GNN** | **KNN** | **GNN** | **KNN** |
| 20% | 0.4201 | **0.5481** | 0.5312 | **0.6630** |
| 10% | 0.3692 | **0.5078** | 0.4519 | **0.6211** |

$\epsilon = 8/255$ on the CIFAR-10 (Krizhevsky et al., 2009) dataset. We use the same training hyperparameters as (Wang et al., 2023b). In each case, we prune 25% of the data. Table 4 summarizes the robustness for the different methods. Score extrapolation considerably improves upon random pruning in terms of robustness and accuracy on clean data. Moreover, it performs only slightly worse than using ground truth scores directly. The KNN extrapolation achieves $0.54$ linear correlation with the ground truth scores, which could be improved by more sophisticated extrapolation approaches.

In Table 5, we additionally provide results for KNN-based extrapolation for a larger dataset and evaluate our approach for $\ell_2$-based adversarial training ($\epsilon = 128/255$). We extrapolate scores from the DU scores from a standard CIFAR-10 training run to 2 million synthetic CIFAR-10 images from (Wang et al., 2023b) and prune 50% of the data. We do not compare to pruning with ground truth scores, as performing a full adversarial training run with 2 million data samples was too expensive. Score extrapolation outperforms random pruning in both settings, while introducing only negligible computational overhead ($< 6\%$) and using only 5% of the full dataset size as a subset for extrapolation. This result demonstrates the effectiveness of score extrapolation in adversarial training, for a considerably smaller initial subset of the full dataset (i.e., $m << n$).

Table 4: Comparison of the utility of original and extrapolated DU scores on the CIFAR-10 dataset for $\ell_\infty$-norm adversarial training ($\epsilon = 8/255$). 25% of the samples are pruned for each method. Random pruning is provided as a baseline.

| Experiment | Robust | Clean |
|---|---|---|
| Random | 47.95% | 80.43% |
| Extrapolated-KNN | 50.39% | 80.86% |
| Ground Truth Scores | 52.26% | 81.95% |

Table 5: Evaluation of extrapolated DU scores on a 2 million sample synthetic CIFAR-10 dataset. 50% of the 2 million synthetic samples are pruned for each method. We provide random pruning as a baseline.

| NORM | EXPERIMENT | ROBUST | CLEAN |
|---|---|---|---|
| $\ell_2$ | RANDOM | 80.79% | 94.20% |
| | EXTRAPOLATED KNN | 81.28% | 94.22% |
| $\ell_\infty$ | RANDOM | 63.13% | 90.86% |
| | EXTRAPOLATED | **63.56%** | 90.50% |

**Ablations.** We support our findings on the concept of score extrapolation with further ablations:

*Initial Subset Sampling.* In Appendix C.1, we investigate the influence of different initial subset selection schemes. The results support the strong performance of i.i.d. (Robert & Casella, 2004) compared to other more balanced but costlier methods.

*Comparison to Dynamic Pruning.* In Appendix C.7, we discuss the differences and similarities to dynamic pruning approaches. Unlike other speed-up approaches, our extrapolation does not alter the training process, making it more flexible while achieving similar time savings and increased accuracy.

*Comparison to Interpolation and Naive Extrapolations.* In Appendix C.6, we compare our extrapolation with a naive score interpolation, a uniform score sampling, and a Multilayer Perceptron without neighborhood information, demonstrating the effectiveness of our suggested approaches.

*Neighborhood Size Analysis.* In Appendices C.3 and C.4 we ablate the influence of different neighborhood sizes and model variants on correlation and task performance.

**Investigating Failures in Extrapolated Scores.** To further understand the properties and limitations of our newly introduced extrapolation paradigm, we performed an analysis of the distribution properties focusing on extrapolated DU scores on the ImageNet in Appendix C.2. It can be seen

that the extrapolation method achieves a moderate correlation between ground truth and estimated scores, but cannot capture the bimodal structure present in the original distribution. This degradation of approximation quality is to be expected, since the distribution is captured by the initially sampled set and the simplified extrapolation between them. We further investigate this behaviour by analyzing samples with high and low extrapolation error. This investigation suggests that high rank differences in predicted importance scores correspond to atypical or out-of-distribution samples—such as those with unusual backgrounds, multiple subjects, or low visual quality.

**Limitations.** In this work, we examined the feasibility of score extrapolation to address the efficiency problem of current data pruning methods. We demonstrated that score extrapolation can improve efficiency for different pruning tasks, scores, and datasets, even with simple extrapolation methods. *Oversmoothing:* However, our initial extrapolation approaches in this work are unable to capture the full distributional complexity of the ground truth scores. This leads to an oversmoothing of the underlying distribution. One solution would be to increase the subset size, but this directly increases costs.
*Extrapolation Methods Complexity:* Another approach to improve the distribution approximation quality is to use more advanced extrapolation networks. Since we focus on the general concept of extrapolation rather than providing a pruning method, we leave this challenge to future work.
*Dataset Size:* Moreover, while our experiments are limited to comparatively small datasets (millions of samples), we believe the potential of extrapolation is even higher for truly large datasets (billions of samples), where subsets can be orders of magnitude smaller yet still sufficient to train an effective extrapolation model.

**Discussion.** Nonetheless, despite the limitations, we could show that our extrapolation concept works reliably. We affirmatively answer the research question *Can importance scores for unseen samples be efficiently extrapolated from a small subset of known scores?*. In addition, using the two provided simple approaches, we demonstrate the effectiveness of three different tasks, showing only a minimal decrease in task performance while providing significant speedups in various settings. While the provision of highly effective approaches remains for future work, we suggest using our KNN-based approach for time constraint settings and our GNN-based approach for more performance-focused applications. Additionally, in contrast to other approaches, our extrapolation *can operate with labels from the initial set only and does not interfere with the training scheme or score calculation*.

## 5 CONCLUSION

The majority of pruning approaches require full training on the dataset to estimate the scores for a subset selection, removing any potential benefit they might provide in most practical applications. In our work, we examined the research question *"Can importance scores for unseen samples be efficiently extrapolated from a small subset of known scores?"*, which addresses the time efficiency problem of the current approaches. With this framework, we select a small subset to estimate the initial scores, resolving the requirement of training on the full set. To mitigate this problem, we propose a novel score extrapolation paradigm. Instead of training on the full set, we select a small subset to estimate the initial scores for the chosen data pruning method and extrapolate these scores to the remaining samples in the dataset using a computationally efficient extrapolation method. Therefore, it is independent of specific training schemes or scoring methods, making them generally applicable. Our experiments with KNN and GNN-based extrapolation approaches show that the extrapolated scores show high correlation with the original scores and achieve a high downstream task performance across two different pruning scores, three different tasks, and four different datasets. Our extrapolation approach achieves speedups of up to $4.9\times$, offering a Pareto-optimal trade-off even compared to training-free pruning methods.

**Outlook.** In future work, we aim to improve downstream task performance, which could include improving extrapolation accuracy, for instance, by refining the GNN-based score estimator by mitigating oversmoothing through residual connections (Scholkemper et al., 2025), exploring further extrapolation methods, and investigating alternative subset selection strategies. Moreover, we aim to extend our extrapolation approach to other data selection tasks involving costly score computations, such as influence functions (Koh & Liang, 2017) and data attribution (Ilyas et al., 2022), but also tasks beyond data selection, such as out-of-distribution detection.

**LLM Statement.** We used LLMs for rephrasing and language checking within the manuscript.

**Reproducibility Statement.** We provided the source code of our experiments and report additionally all parameters and experiment settings in Appendix A. We used multiple seeds to allow statistical significance.

**Ethics Statement.** Our work addresses the field of data subset selection. Data subset selection is a fundamental part of many machine learning applications. We acknowledge that data subset selection can also be applied to train a model to cause harm. Yet, we propose a novel extrapolation paradigm to resolve the inherent time efficiency problem, and we do not see any ethical concerns.

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

# A EXPERIMENT SETUP

## A.1 DATASET

To evaluate the efficacy of our proposed score extrapolation framework, we conduct experiments on four image classification datasets differing in scale, number of classes, and image resolutions: CIFAR-10 (Krizhevsky et al., 2009) (50K samples, 10 classes, $32 \times 32$), synthetic CIFAR-100 (Wang et al., 2023b) (1M samples, 100 classes, $32 \times 32$, generated with denoising diffusion models (Ho et al., 2020)), Places-365 (Zhou et al., 2017) (1.80M samples, 365 classes, resized to $64 \times 64$ to expedite experiments), and Imagenet-1k (Deng et al., 2009) (1.28M samples, $1000$ classes, downscaled version of $64 \times 64$). To evaluate model accuracy after pruning, we use the original test sets of each dataset, with the exception of synthetic CIFAR-100, for which we employ the standard CIFAR-100 (Krizhevsky et al., 2009) test set. Additionally, we explore the unsupervised dataset pruning with standard CIFAR-10, and adversarial training with synthetic CIFAR-10 (Wang et al., 2023b) (2M samples).

**Synthetic CIFAR-100:** To evaluate our extrapolation approaches on large-scale settings, where many methods incur non-linear computational costs proportional to $n$, we make use of the Synthetic CIFAR-100 dataset[2] (Wang et al., 2023b). This dataset comprises several million-scale variants of CIFAR-100 generated using an Elucidated Diffusion Model (EDM) (Karras et al., 2022), resulting in high-fidelity samples with strong FID scores. As a result, the synthetic images can be used for model training and for sim-to-real transfer without requiring additional real CIFAR-100 data.

The use of this dataset allows us to study performance at a substantially larger number of samples $n$ while preserving the semantic structure and distributional properties of CIFAR-100. Moreover, it provides a more realistic ratio of samples per class for large-scale scenarios, reflecting the imbalance often seen in "internet-scale" datasets, where the number of classes is fixed while data volume grows dramatically.

**Dataset License:** All datasets used in our experiments are publicly available, and most of them are widely used in the ML community. The standard CIFAR-10 and CIFAR-100 (Krizhevsky et al., 2009) datasets are freely available for research and educational purposes without any licensing requirements. ImageNet (Deng et al., 2009) is available for free to researchers for non-commercial use, but does not outline a specific license. Both synthetic CIFAR-10 and synthetic CIFAR-100 (Wang et al., 2023b) are publicly available under the MIT license. Similarly, Places365 (Zhou et al., 2017) is released under the MIT license. We performed experiments adhering to the licensing terms of the respective datasets.

## A.2 STATISTICAL SIGNIFICANCE

Since the computed scores depend on the training, which is stochastic by nature, the scores obtained at the end are also stochastic. To ensure statistical robustness, for any dataset, we compute three different sets of ground truth scores $S \in \mathbb{R}^n$ with three different random initializations. For each set of scores, we compute the model accuracy at various pruning rates: $10, 20, 50, 80, 90$ and $95$ percentages. To compute subset scores $S_s$, we follow the same procedure. We randomly select $\mathbb{D}_s \subset \mathbb{D}$ of cardinality $m$, and compute the scores $S_s$ with standard pruning algorithms. We do this three times with three different seeds. For each set of $S_s$, we extrapolate using the KNN and GNN approaches. In this way, we obtain six sets of extrapolated scores $S_r$, (three from KNN, and three from GNN). Note that extrapolation with the GNN itself is stochastic in nature. For each set of scores, we prune the data at various rates and train the model on the pruned dataset once. Thus, for each pruning rate, we get three test accuracy values for both extrapolation methods. Figure 2, Figure 8 and Figure 3 report this mean accuracy.

## A.3 COMPUTATIONAL RESOURCES

All experiments are conducted using NVIDIA A100-PCIE GPUs, with 42.4 GB of VRAM. Computational time reported in Table 1, and Figure 3 are the total mean runtime (in minutes) required to

---

[2]https://github.com/wzekai99/DM-Improves-AT

compute $S_s$, extrapolate, and subsequent model training on the pruned dataset. Time for ground truth scores reflects the mean time required for full dataset scoring plus training on the pruned subset.

### A.4 MODELS

Both DU, and TDDS require model training for numerous epochs to compute the scores. To validate that score extrapolation works with different models $\mathcal{F}_s$, we used ResNet-18 (He et al., 2016) for Cifar-10, and Imagenet, and ResNet-50 for Synthetic CIFAR-100, and Places-365. For the adversarial setting, we used Wide-ResNet-28-10 (Zagoruyko & Komodakis, 2016). During extrapolation, samples are represented in the embedding space induced by these models.

For the unsupervised setting, we employ DINOv2 (Oquab et al., 2023) as a foundation model to obtain fixed embeddings for all samples. Both extrapolation procedures (KNN-based and GNN-based) are subsequently performed in the embedding space of this foundation model. This diversity of architectures and training paradigms demonstrates that our extrapolation approach is not restricted to a specific model but can be applied broadly across a range of backbone networks and training schemes.

### A.5 EXPERIMENT HYPERPARAMETERS

We collected all hyperparameter settings in Tables 6 to 9. They are properly introduced with the scored description in the following section.

### A.6 TEMPORAL ANALYSIS

In this subsection, we clarify the temporal complexity of our extrapolation pipeline, provide an explicit time–cost derivation, and discuss practical concerns and limitations.

**Temporal Structure of Extrapolation and Regular Pruning.**

Given the definition from Section 3, we have the full dataset $\mathbb{D}$ and the randomly i.i.d. sampled subset $\mathbb{D}_s \subset \mathbb{D}$, the size of size $m$, for which we use 10-20% of the size of $\mathbb{D}$, defined as the fractional size $\alpha$. Since the sampling is random, the costs are negligible. For 10% $\alpha = 0.1$. We decompose the total cost into the following components:

**Score Extrapolation Pruning.**

- $T_{\text{train}}(\alpha)$: Cost of training the task model on an $\alpha$-fraction of the data.
- $T_{\text{score}}(\alpha)$: Cost of computing the ground truth pruning scores on an $\alpha$-fraction of the data.
- $T_{\text{embed}}$: Cost of obtaining embeddings for all samples (one forward pass each).
- $T_{\text{extra}}$: Cost of score extrapolation via either KNN or GNN.
- $T_{\text{prune}}$: Final training on the pruned subset for evaluation, shared across all methods.

Such that the total costs can be estimated as:

$$T_{\text{extra}} = T_{\text{train}}(\alpha) + T_{\text{score}}(\alpha) + T_{\text{embed}} + T_{\text{extra}} + T_{\text{prune}}. \tag{8}$$

**Standard score-based pruning.**

- $T_{\text{train}}(1.0)$: Cost of training the task model on the full dataset.
- $T_{\text{score}}(1.0)$: Cost of computing pruning scores on the full dataset.
- $T_{\text{prune}}$: Final training on the pruned subset for evaluation, shared across all methods.

$$T_{\text{standard}} = T_{\text{train}}(1.0) + T_{\text{score}}(1.0) + T_{\text{prune}}. \tag{9}$$

Since the cost of final training on $T_{\text{prune}}$ is identical for both approaches, the remaining terms govern the comparison. A temporal benefit is achieved if the following inequality is fulfilled:

$$T_{\text{train}}(s) + T_{\text{score}}(s) + T_{\text{embed}} + T_{\text{extra}} < T_{\text{train}}(1.0) + T_{\text{score}}(1.0),$$

.

**Why Extrapolation Is Cheaper.**

*Training and Scoring:* The dominant runtime difference arises from the first training phase: training on 100% of the data ($T_{\text{train}}(1.0)$) for multiple epochs is substantially more expensive than training on 10–20% of the data ($T_{\text{train}}(\alpha)$), even when including full embedding and extrapolation.

*Embeddings:* Creating embeddings on the entire dataset ($T_{\text{embed}}$) requires exactly one forward pass per sample,

$$T_{\text{embed}} \approx |\mathbb{D}| \cdot T_{\text{fwd}},$$

while training requires both forward and backward passes for $E$ epochs,

$$T_{\text{train}}(1.0) \approx E \cdot |\mathbb{D}| \cdot (T_{\text{fwd}} + T_{\text{bwd}}).$$

Assuming $T_{\text{bwd}} \approx T_{\text{fwd}}$, one training epoch costs approximately twice an embedding pass. Thus, the creation of the embeddings is cheaper than training one epoch for a subset size of $<= 50\%$. For common settings (10-20% subset size compared to 50-200 epochs for training), the embedding cost is orders of magnitude smaller than full training.

*Extrapolation:* Given $n$ target samples and feature dimension $d$, KNN extrapolation costs $O(nd)$, which is negligible relative to model training. This is reflected empirically in our Pareto plots.

GNN models operate on the embedding graph and contain far fewer parameters than the task model: A ResNet 50 has 23,508,032 trainable parameters, while the GNN we used has 1,367,553 parameters. In addition, the training requires significantly fewer iterations than end-to-end supervised training, resulting in a lower overall cost.

**Quantitative Analysis.** Since the exact number of epochs and dimension can differ from dataset to dataset, we performed a time-vs-accuracy analysis in Section 4 in Figure 3, taking *all step* into account. The experiments demonstrate that our approach is consistently a temporally cheaper and Pareto-optimal compared to standard pruning. Additionally, extrapolation maintains competitive downstream accuracy.

## B SCORES

We assess our extrapolation framework with two state-of-the-art dataset pruning methods: Dynamic Uncertainty (DU) (He et al., 2024), and Temporal Dual-Depth Scoring (TDDS) (Zhang et al., 2024).

### B.1 DU SCORES

Given a model $\theta_k$ trained over $K$ epochs, $U_k(x)$ for a sample $x$ at epoch $k$ is computed as the standard deviation of the predicted probabilities $\mathbb{P}(y|x, \theta_k)$ over a sliding window of $J$ epochs (He et al., 2024):

$$S_k(x) = \sqrt{\frac{1}{J-1} \sum_{j=0}^{J-1} \left[ \mathbb{P}(y|x, \theta_{k+j}) - \bar{\mathbb{P}} \right]^2},$$

where $\bar{\mathbb{P}} = \frac{1}{J} \sum_{j=0}^{J-1} \mathbb{P}(y|x, \theta_{k+j})$. The final dynamic uncertainty score $S(x)$ for each sample is computed by averaging over all sliding windows:

$$S(x) = \frac{1}{K-J} \sum_{k=0}^{K-J-1} S_k(x),$$

For experiments, we set $J = 10$, and $K = 50$ for CIFAR-10, synthetic CIFAR-100 and PLACES-365, while $K = 90$ is used for Imagenet. More details on the hyperparameters are provided in Table 6.

## B.2 TDDS Scores

TDDS (Zhang et al., 2024) computes the importance score for a sample $x$ by quantifying its contribution to optimization dynamics. Specifically, TDDS calculates the epoch-wise change in loss, $\Delta \ell_k(x)$, projected onto the model's optimization trajectory. Formally, for a sliding window of size $K$, the score is computed as:

$$S(x) = \sum_{k=J}^{K} \beta(1-\beta)^{K-k} \sum_{j=k-J+1}^{k} \left( |\Delta \ell_j(x)| - \frac{1}{J} \sum_{i=k-J+1}^{k} |\Delta \ell_i(x)| \right)^2,$$

where $\Delta \ell_k(x)$ measures the KL-divergence of predictions between consecutive epochs:

$$\Delta \ell_k(x) = f_{\theta_{k+1}}(x)^\top \log \frac{f_{\theta_{k+1}}(x)}{f_{\theta_k}(x)},$$

and $\beta$ is an exponential decay factor. In experiments, we set $J = 10$, and $\beta = 0.9$ for all datasets, and $K = 50$ for CIFAR-10, synthetic CIFAR-100, and Places-365, whereas $K = 90$ for ImageNet. Further details are provided in Table 7.

## B.3 Unsupervised DU Scores

To assess whether our score-extrapolation framework remains effective in the absence of ground-truth labels, we employ TURTLE (Gadetsky et al., 2024). TURTLE assigns a pseudo-label to each sample $x \in \mathcal{D}$ by optimizing a bilevel objective within the representation space induced by a foundation model $\phi$.

During TURTLE optimization, at each outer iteration $k \in \{1, \ldots, K\}$, we record the softmax probability vector:

$$\mathbf{p}^{(k)}(x) = \text{softmax}\left( \boldsymbol{\theta}^{(k)} \phi(x) \right) \in \Delta^{C-1},$$

where $\boldsymbol{\theta}^{(k)}$ denotes the learnable linear transformation at iteration $k$, and $C$ is the number of classes specified a priori. After $K$ outer iterations, we define the final pseudo-label for $x$ as

$$\hat{y}(x) = \arg\max_{c \in \{1, \ldots, C\}} \mathbf{p}_c^{(K)}(x).$$

Analogous to the supervised setting, we perform post-hoc computation to obtain the uncertainty at the pseudo-label $\hat{y}(x)$ across a sliding window of length $J$ over epochs and subsequently average these values to compute the *unsupervised-DU* score for each sample.

Similar to the supervised setting, we use $J = 10$ for the experiments. Further hyperparameter details are provided in Table 9, which follows settings in (Zhang et al., 2024).

While the dynamic uncertainty estimation works sufficiently, its effectiveness for unsupervised data pruning is limited. Since this unsupervised importance score-based pruning is uncommon, we focus on the related experiments on the correlation between orginal and extrapolated scores.

## B.4 Scores Extrapolation

For KNN-based extrapolation, we computed the $k$ nearest neighbors using the Euclidean distance. To assess how the choice of $k$ affects extrapolation, we varied $k$ across $10, 20, 50, 100$ and evaluated the correlation between the extrapolated and ground-truth scores (based on $S$) for samples in $\mathbb{D}_r$. The value of $k$, yielding the highest Pearson correlation, is reported in our main results (Table 2), while the full ablation is presented in Table 11.

For GNN-based extrapolation, we similarly examined the effect of the neighborhood size $k$ while constructing the graph. We used different values of $k$ ($10, 20,$ and $50$). The GNN comprises three GCN (Kipf & Welling, 2017) layers (hidden dimensions $512$ and $256$) and an output layer producing scalar importance scores. We use dropout regularization of $0.5$ to avoid overfitting. To ensure

Table 6: Hyperparameters and experimental settings for all datasets to compute standard DU scores $\mathbb{S}$, as well as subset DU scores $\mathbb{S}_s$. Subset sizes are reported as a percentage of the total dataset size. To compute standard scores, training is done on the complete dataset

| Hyperparameters | CIFAR-10 | Synthetic CIFAR-100 | Places-365 | ImageNet |
|---|---|---|---|---|
| Num epochs ($K$) | 50 | 50 | 50 | 90 |
| Batch size ($B$) | 256 | 256 | 128 | 256 |
| Model ($\mathcal{F}$) | ResNet-18 He et al. (2016) | ResNet-50 | ResNet-50 | ResNet-18 |
| Optimizer | Adam Kingma & Ba (2015) | Adam | Adam | Adam |
| Learning rate ($\eta$) | $10^{-3}$ | $10^{-3}$ | $10^{-3}$ | $10^{-3}$ |
| Weight decay ($\lambda$) | $10^{-4}$ | $10^{-4}$ | $10^{-4}$ | $10^{-4}$ |
| Scheduler | None | None | None | None |
| Window ($J$) | 10 | 10 | 10 | 10 |
| Subset size ($m$) | 40%, 20% | 30%, 20%, 10% | 20%, 10% | 20%, 10% |

Table 7: Hyperparameters to compute standard and subset TDDS scores.

| Hyperparameters | CIFAR-10 | Synthetic CIFAR-100 | Places-365 |
|---|---|---|---|
| Num epochs ($K$) | 50 | 50 | 50 |
| Batch size ($B$) | 256 | 256 | 128 |
| Model ($\mathcal{F}$) | ResNet-18 | ResNet-50 | ResNet-50 |
| Optimizer | SGD | SGD | SGD |
| Learning rate ($\eta$) | $10^{-3}$ | $10^{-3}$ | $10^{-3}$ |
| Weight decay ($\lambda$) | $5 \times 10^{-4}$ | $5 \times 10^{-4}$ | $5 \times 10^{-4}$ |
| Momentum | 0.9 | 0.9 | 0.9 |
| Nesterov Sutskever et al. (2013) | True | True | True |
| Scheduler | CosineAnnealing | CosineAnnealing | CosineAnnealing |
| Window ($J$) | 10 | 10 | 10 |
| Trajectory | 10 | 10 | 10 |
| Exponential decay ($\beta$) | 0.9 | 0.9 | 0.9 |
| Subset size ($m$) | 40%, 20% | 20%, 10% | 20%, 10% |

scalability on a large dataset, we use neighbor sampling (Hamilton et al., 2017), with mini-batches of node size 128.

We randomly split $m$ samples in $\mathbb{D}_s$ into 90% training set and 10% validation set. These nodes already have the computed scores $S_s$. We train GNN for 25 epochs, and the checkpoint achieving the highest Pearson correlation between predicted and reference scores (based on $S_s$) on the validation set is selected for inference. Scores for all samples in $\mathbb{D}_r = \mathbb{D} \setminus \mathbb{D}_s$ are inferred using this model checkpoint. We report the correlation between the inferred scores and scores with the standard approach ($S$) for the samples in $\mathbb{D}_r$.

Interestingly, we observe that GNNs usually achieved the best performance with smaller neighborhood sizes ($k = 10$), suggesting that message passing enables effective propagation of information even with sparse local connectivity. Detailed results are provided in Table 12, with the best-performing configuration reported in Table 2.

Further, we examined the influence of labels on the performance of the GNN extrapolator when provided as node embedding. We performed an ablation study comparing variants trained with and without one-hot encoded class labels appended to the node features. Extrapolation quality was measured using the Pearson correlation between the models' predicted scores and the oracle scores on the unscored set $\mathbb{D}_r$. Incorporating class labels consistently improved performance across datasets and score types, demonstrating that the semantic class information provides a useful inductive bias for guiding extrapolation. As shown in Table 10, we observe relative gains of +6.8% on ImageNet (DU), +7.9% on Places365 (DU), and +4.5% on Places365 (TDDS).

Table 8: Hyperparameters used for training models on pruned datasets. Both random pruning, and score based pruning (standard as well as extrapolated scores) use the same configurations

| Hyperparameters | CIFAR-10 | Synthetic CIFAR-100 | Places-365 | ImageNet |
|---|---|---|---|---|
| Num epochs ($K$) | 50 | 50 | 50 | 90 |
| Batch size ($B$) | 256 | 256 | 128 | 256 |
| Model ($\mathcal{F}$) | ResNet-18 | ResNet-50 | ResNet-50 | ResNet-18 |
| Optimizer | Adam | Adam | Adam | Adam |
| Learning rate ($\eta$) | $10^{-3}$ | $10^{-3}$ | $10^{-3}$ | $10^{-3}$ |
| Weight decay ($\lambda$) | $10^{-4}$ | $10^{-4}$ | $10^{-4}$ | $10^{-4}$ |
| Scheduler | OneCycle | OneCycle | OneCycle | OneCycle |
| Window ($J$) | 10 | 10 | 10 | 10 |

Table 9: Hyperparameters to compute unsupervised DU scores (for full dataset, as well as subset)

| Hyperparameters | CIFAR-10 |
|---|---|
| Num epochs ($K$) | 400 |
| Batch size ($B$) | 10,000 |
| Inner steps ($M$) | 10 |
| Representation ($\phi$) | DINOv2 Oquab et al. (2023) |
| Optimizer | Adam |
| Regularization coeff. ($\gamma$) | 10 |
| Inner Learning rate ($\eta_{\text{in}}$) | $10^{-3}$ |
| Outer Learning rate ($\eta_{\text{out}}$) | $10^{-3}$ |
| Weight decay ($\lambda$) | $10^{-3}$ |
| Scheduler | None |
| Window ($J$) | 10 |
| Subset size ($m$) | 40% |

## C  ADDITIONAL EXPERIMENTS

### C.1  INITAL SUBSET SELECTION

In our framework, we use random sampling to create an initial subset for training. While we assume little prior knowledge about our dataset, Uniform/IID sampling provides an unbiased estimate of the underlying data distribution and converges to the true distribution as the number of samples increases (Robert & Casella, 2004). Similarly, classical results in statistical learning theory—such as PAC bounds and uniform convergence—rely on the IID assumption to ensure that empirical estimates generalize reliably to the true distribution (Shalev-Shwartz & Ben-David, 2014). To ensure that our subset provides an unbiased estimate of the properties of the full dataset, IID selection is therefore a natural choice. Informed subset selection strategies (e.g., coreset construction or uncertainty-based sampling) may introduce distributional biases that underrepresent certain regions of the input space. For example, samples that we would like to prune later on also need to be represented in the subset, which may not be the case for informed algorithms (Settles, 2010). Nonetheless, introducing geometric approaches like CoreSet (Sener & Savarese, 2018) or ZCoreSet (Griffin et al., 2024) adds computation overhead.

We provide additional results for further sampling algorithms in the Table 13. It can be seen that random and stratified sampling provide the highest correlation. Zcoreset requires further computation and knowledge about the data, being relatively expensive. Using Zcoreset with a foundation model on Places365 with 95% pruning, our KNN approach takes 3538 seconds compared to ZCoreSet's 137920 seconds.

Additionally, we investigate our embedding space assumption to examine the coverage and representativeness of the embedding space.

Table 10: Ablation study evaluating the effect of concatenating one-hot class labels to node features on extrapolation performance. Reported values are Pearson correlations on the unscored set scores $S_r$ for a neighborhood size of $k = 10$.

| Dataset | Score Type | Without Labels | With Labels | Improvement |
|---|---|---|---|---|
| ImageNet | DU | 0.3925 | **0.4193** | +6.8% |
| Places365 | DU | 0.3710 | **0.4004** | +7.9% |
| Places365 | TDDS | 0.2518 | **0.2632** | +4.5% |

Table 11: Pearson and Spearman correlations for different $k$ in KNN pruning methods across datasets.

| Dataset | Method | Sample size (%) | Pearson $\rho$ | | | | Spearman $r_s$ | | | |
|---|---|---|---|---|---|---|---|---|---|---|
| | | | $k{=}10$ | $k{=}20$ | $k{=}50$ | $k{=}100$ | $k{=}10$ | $k{=}20$ | $k{=}50$ | $k{=}100$ |
| Imagenet | DU | 20 | 0.3764 | **0.3779** | 0.3698 | 0.3607 | 0.2911 | **0.3068** | 0.3057 | 0.2982 |
| | | 10 | 0.2507 | 0.2572 | **0.2575** | 0.2513 | 0.1773 | 0.1888 | **0.1980** | 0.1894 |
| Places365 | DU | 20 | 0.2904 | 0.3054 | **0.3081** | 0.3004 | 0.2283 | 0.2446 | **0.2524** | 0.2487 |
| | | 10 | 0.2139 | 0.2196 | **0.2215** | 0.2184 | 0.1687 | 0.1722 | **0.1791** | 0.1715 |
| | TDDS | 20 | 0.2169 | 0.2208 | **0.2251** | 0.2213 | 0.2128 | 0.2180 | **0.2214** | 0.2159 |
| | | 10 | 0.1497 | 0.1561 | **0.1620** | 0.1583 | 0.1438 | 0.1526 | **0.1594** | 0.1513 |
| Synthetic | DU | 30 | 0.4964 | 0.5092 | **0.5118** | 0.5106 | 0.6675 | 0.6864 | **0.6962** | 0.6955 |
| | | 20 | 0.4447 | 0.4530 | **0.4538** | 0.4499 | 0.6355 | 0.6493 | **0.6562** | 0.6562 |
| | | 10 | 0.3123 | 0.3197 | **0.3243** | 0.3202 | 0.5286 | 0.5374 | **0.5471** | 0.5438 |
| | TDDS | 20 | 0.3821 | 0.3886 | **0.3955** | 0.3910 | 0.6067 | 0.6122 | **0.6208** | 0.6197 |
| | | 10 | 0.3096 | 0.3178 | **0.3273** | 0.3147 | 0.5137 | 0.5261 | **0.5324** | 0.5311 |
| CIFAR-10 | DU | 40 | 0.6149 | 0.6328 | **0.6371** | 0.6313 | 0.6225 | 0.6411 | **0.6477** | 0.6447 |
| | | 20 | 0.2721 | 0.2759 | **0.2765** | 0.2762 | **0.2603** | 0.2597 | 0.2558 | 0.2507 |
| | TDDS | 40 | 0.3841 | 0.4126 | **0.4181** | 0.4158 | 0.5593 | 0.5898 | 0.6029 | **0.6040** |
| | | 20 | 0.2014 | 0.2128 | **0.2134** | 0.2038 | 0.2738 | 0.2806 | **0.2764** | 0.2611 |

To examine distributional coverage, we computed Fréchet Inception Distance (FID) scores between sampled subsets (20%) and the remaining training data across different sampling methods in Table 14.

To put these scores into context: FID scores around 2 indicate strong distributional similarity for CIFAR10 (Wang et al., 2023a). Similarly, the FID between high-quality synthetic CIFAR100 data (Wang et al., 2023a) and the real dataset is 2.74. These results provide additional quantitative support that our sampled subsets offer good coverage of the data distribution. They also suggest that biased selection methods, such as Zcoreset, may yield poorer approximations.

## C.2 LIMIATION ANALYSIS

**Visual Analysis** In Figure 6 (a), we perform a deeper investigation regarding limitations of our extrapolation approach, superficially focusing on extrapolated DU scores on the ImageNet dataset. While the extrapolation method achieves a moderate correlation between ground truth and estimated scores, the extrapolated scores fail to capture the bimodal structure present in the original distribution, instead forming a narrower, unimodal distribution with a higher mean and lower variance. This oversmoothing likely contributes to the observed extrapolation errors.

To further investigate these errors, we highlight samples with high and low rank differences (specifically using one "dog" class, but other classes showed similar patterns). Qualitative examples in subfigures (b) and (c) suggest that high rank differences often correspond to atypical or out-of-distribution samples—such as those with unusual backgrounds, multiple subjects, or low visual quality—whereas low rank difference samples tend to be prototypical, centered, and consistent in appearance. This indicates that the extrapolation method struggles most with outliers and visually ambiguous inputs. We hypothesize that more powerful extrapolation methods or using more seed data to train the score extrapolation would improve score extrapolation in these cases.

Table 12: Pearson and Spearman correlations for different $k$ in GNN based extrapolation

| Dataset | Method | Sample size (%) | Pearson $\rho$ | | | Spearman $r_s$ | | |
|---|---|---|---|---|---|---|---|---|
| | | | $k=10$ | $k=20$ | $k=50$ | $k=10$ | $k=20$ | $k=50$ |
| Imagenet | DU | 20 | **0.4193** | 0.4139 | 0.4045 | **0.3503** | 0.3442 | 0.3388 |
| | | 10 | **0.2850** | 0.2734 | 0.2784 | 0.2178 | **0.2189** | 0.2159 |
| Places365 | DU | 20 | **0.4004** | 0.3869 | 0.3798 | **0.3608** | 0.3443 | 0.3373 |
| | | 10 | **0.2612** | 0.2604 | 0.2577 | **0.2158** | 0.2140 | 0.2095 |
| | TDDS | 20 | **0.2632** | 0.2615 | 0.2557 | **0.2646** | 0.2611 | 0.2559 |
| | | 10 | **0.2372** | 0.2275 | 0.2219 | **0.2297** | 0.2238 | 0.2184 |
| Synthetic | DU | 30 | 0.5593 | 0.5606 | **0.5634** | 0.7300 | **0.7344** | 0.7281 |
| | | 20 | **0.4910** | 0.4886 | 0.4829 | **0.7009** | 0.6983 | 0.6942 |
| | | 10 | **0.3396** | 0.3320 | 0.3282 | **0.5593** | 0.5513 | 0.5472 |
| | TDDS | 20 | **0.4236** | 0.4193 | 0.4155 | **0.6713** | 0.6711 | 0.6673 |
| | | 10 | **0.3849** | 0.3801 | 0.3763 | **0.5722** | 0.5715 | 0.5639 |
| CIFAR-10 | DU | 40 | 0.6163 | **0.6318** | 0.6197 | 0.6533 | **0.6608** | 0.6582 |
| | | 20 | **0.4080** | 0.2853 | 0.2574 | **0.3915** | 0.3009 | 0.2619 |
| | TDDS | 40 | **0.3891** | 0.3217 | 0.3007 | **0.5656** | 0.4213 | 0.4076 |
| | | 20 | **0.1763** | 0.1630 | 0.1571 | **0.2008** | 0.1120 | 0.1095 |

Table 13: Correlation coefficients for different sampling methods.

| Sampling Method | Pearson Correlation | Spearman Correlation |
|---|---|---|
| Random | 0.4004 | 0.3611 |
| Zcoreset | 0.3852 | 0.3508 |
| Stratified | 0.4028 | 0.3619 |

Table 14: Similarity of different sampling methods across datasets.

| Sampling Method | CIFAR10 | CIFAR100 | Places365 |
|---|---|---|---|
| Random | 2.11 | 0.11 | 0.07 |
| Stratified | 2.11 | 0.11 | 0.07 |
| Zcoreset | 2.42 | 0.14 | 0.10 |

Given the observation of the oversmoothing and the rank difference relating mostly to more atypical samples, we assume issues can be mitigated in two ways: either by increasing the subsampling size for a more diverse dataset to increase the number of points describing the underlying distribution or by increasing the model complexity for modelling the distributions more accurately. An interpretation of the model complexity can be gleaned from Tables 11 and 12, where it is evident that for simpler datasets, such as CIFAR-10, the correlation difference between the KNN and GNN extrapolation is small. In contrast, for larger datasets, the increased model complexity improves the correlation. To validate the first mitigation strategy, we increased the subset size to 60% in Figure 5. It can be seen that the correlation and distribution similarity increase drastically, such that the binomial nature is evident.

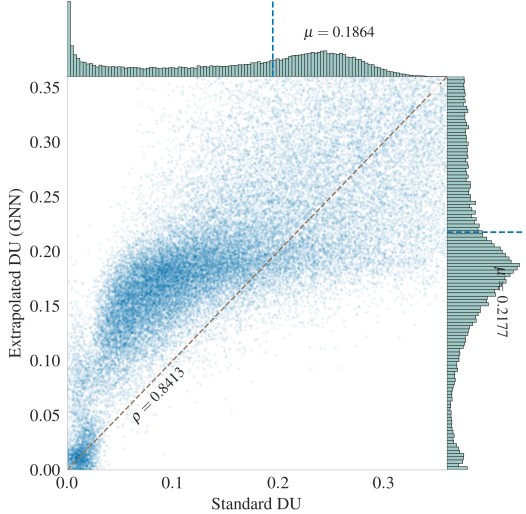

Figure 5: Results using an increased initial subset of 60%, which drastically improves the capturing ability of the distribution. (Format will be updated)

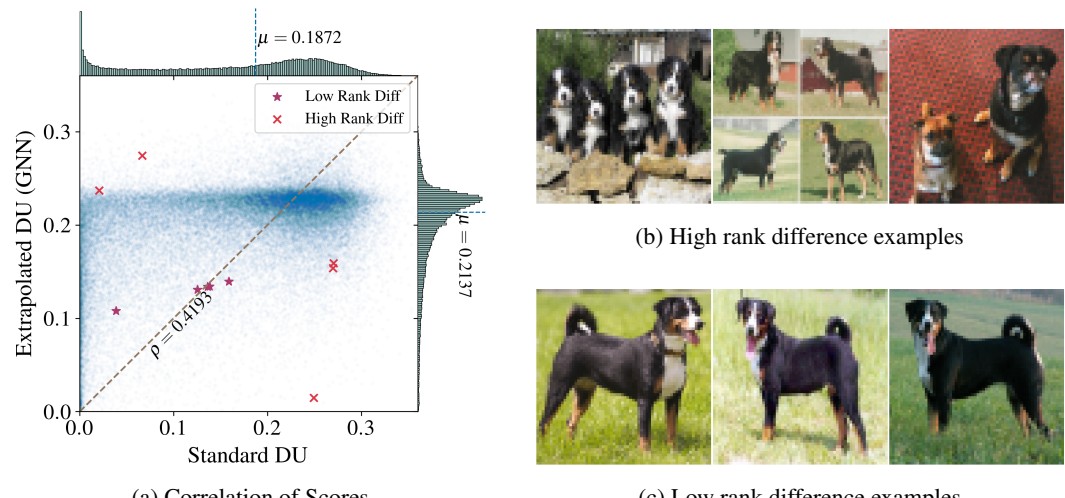

(a) Correlation of Scores         (b) High rank difference examples

                                             (c) Low rank difference examples

Figure 6: Score distribution (a) and qualitative analysis (b-c) of extrapolation errors for ImageNet and DU. a) Extrapolated scores show a moderate correlation with ground truth but miss the bimodal structure, resulting in a narrower, oversmoothed distribution. (b) and (c) show examples with high and low rank discrepancies. High discrepancies often correspond to outliers with atypical backgrounds or multiple objects, while low discrepancies align with prototypical class examples.

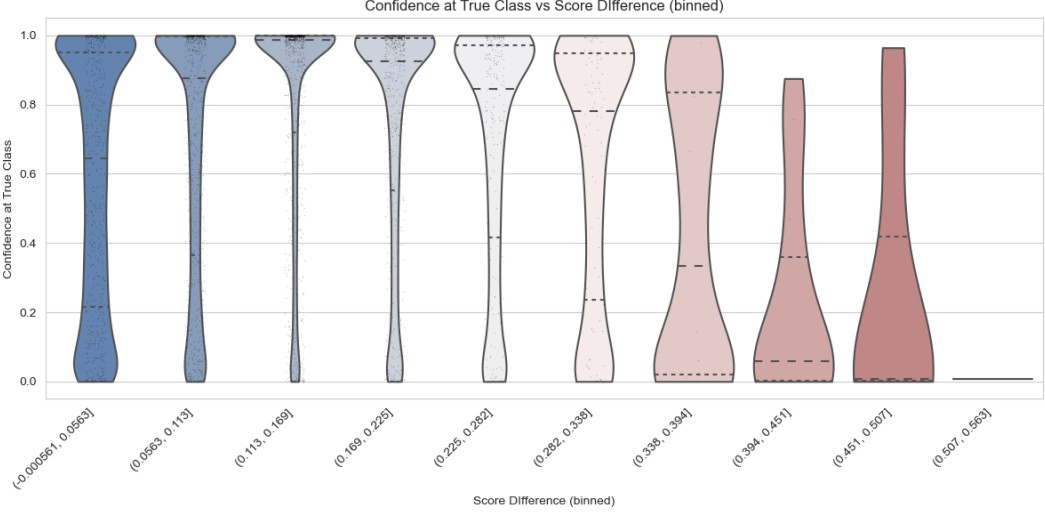

Figure 7: Correlation between confidence for the true class and score difference as a violin plot. It can be seen that high confidences relate to low score differences, while high score differences are mapped to low confidence in the correct class.

**Statistical Analyis:** In addition to visual analy-
sis of the limits of our extrapolation, we investigated the correlation between the differences between predicted and ground-truth scores to further identify failure cases. In Figure 7, we show the correlation between the score differences and the confidence for the true class of the prediction. It can be seen that for sampling with low score differences, the predictions are primarily correct. While increasing score differences, the confidence drops rapidly, and in the last bin, there are only misclassified samples. This shows that the quality of the extrapolation is related to the misprediction rate of a sample, which can be interpreted as a measure of the sample's difficulty.

Table 15: Pearson ($\rho$) and Spearman ($r_s$) correlations of scores correlation between extrapolated and original scores, and post-pruning accuracy ($A$)

| Dataset | Prune % | Method | $\rho(\rho, A)$ | $\rho(r_s, A)$ | $r_s(\rho, A)$ | $r_s(r_s, A)$ |
|---|---|---|---|---|---|---|
| Places365 | 50 | DU | 0.977 | 0.975 | 0.771 | 0.771 |
| | 95 | TDDS | 0.914 | 0.914 | 1.000 | 1.000 |
| ImageNet | 50 | DU | 0.766 | 0.779 | 0.771 | 0.771 |
| Synthetic CIFAR | 95 | DU | 0.995 | 0.944 | 0.943 | 0.943 |
| | | TDDS | 0.901 | 0.940 | 1.000 | 1.000 |

## C.3 Relationship between Correlation and Downstream Task Accuracy

We investigate whether higher Pearson ($\rho$) or Spearman rank ($r_s$) correlations between extrapolated and ground truth scores are indicative of improved pruning performance, particularly in regimes where ground truth pruning outperforms random pruning. For the highest pruning rate at which ground truth pruning yields superior accuracy to random pruning, we compare the downstream accuracies obtained by retaining top-scoring samples according to various scoring methods (extrapolated, standard, and random). Note that random pruning corresponds to 0 correlation, while ground truth scores correspond to perfect correlation ($\rho = 1$, $r_s = 1$). Our findings indicate that increased correlation between extrapolated and ground truth scores leads to downstream accuracies that closely match those of ground truth-based pruning (see Figure 4). We further quantify the relationship between correlation metrics and downstream accuracy ($A$) after pruning by computing the Pearson and Spearman correlations between these variables. Results are summarized in Table 15.

## C.4 Pruning Performance with Smaller Subset Sizes

In Figure 2 we evaluated the pruning performance of Random Pruning, Standard pruning (DU, and TDDS), and extrapolation-based pruning with initial score computation on a subset of size $m = 20\%$ (20% for Places-365, 40% for CIFAR-10). Here we provide additional results examining the impact of reducing the initial subset size, specifically considering $m = 10\%$ (20% for CIFAR-10). The results are presented in Figure 8.

We observe that, consistent with the initial larger subset size (Figure 2), pruning with extrapolated scores, even with a smaller initial subset size, outperforms the random baseline whenever the respective standard scores do. However, as the initial subset size $m$ decreases, the effectiveness of extrapolated scores diminishes. They also have smaller Pearsons correlation and Spearman rank with the standard score, as demonstrated in Tables 2, 11 and 12.

## C.5 Rationale for Score Extrapolation

The core premise of score extrapolation is that semantically similar data points should possess similar importance scores. Importance scores, whether derived from prediction uncertainty, gradient norms, or other training dynamics, are fundamentally tied to how a model perceives and interacts with the data. A model trained to separate classes learns a representation manifold where proximity in the embedding space reflects semantic similarity.

It follows that samples located close to each other in this space will likely elicit similar responses from the model, leading to correlated importance scores. For instance, a dense cluster of prototypical images from the same class represents redundant information and should receive uniformly low importance. Conversely, ambiguous samples or those near a class decision boundary are critical for learning and should receive high importance. Our extrapolation methods are designed to leverage this structure: KNN directly applies a local smoothness prior, while the GNN learns a more complex function still grounded in the geometry and local class composition of the representation manifold.

**Empirical Validation** To empirically validate our hypothesis, we analyzed the relationship between ground-truth DU scores and the local neighborhood structure in the embedding space of a ResNet-50 model trained on ImageNet-1k. For each sample, we computed the Pearson correlation between its

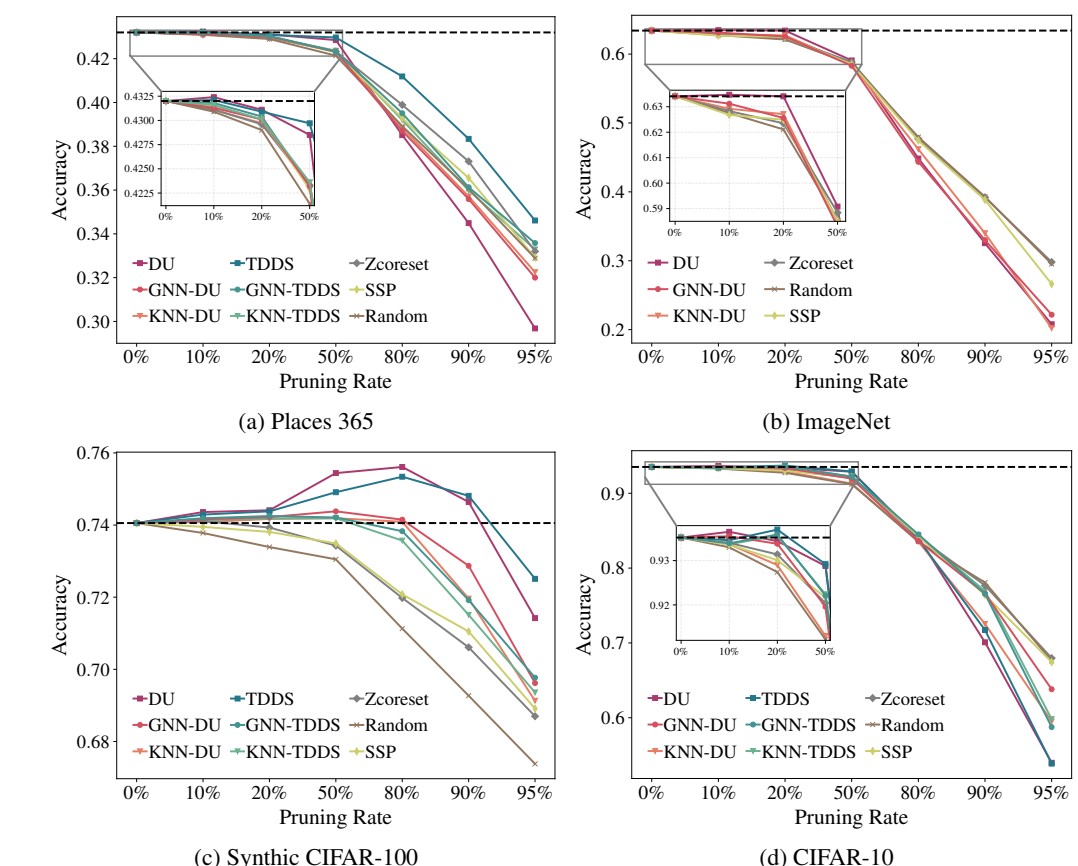

Figure 8: Pruning performance of Standard approaches and their extrapolated counterparts, which started with score computation on 10% subset (20% for CIFAR-10), and then extrapolated with KNN, and GNN.

importance score and several metrics characterizing its 20 nearest neighbors. The results, summarized in Table 16, confirm our intuition.

Table 16: Pearson correlation between ground-truth DU scores and local neighborhood metrics in the ImageNet-1k embedding space ($k = 20$). "Same-label" refers to neighbors sharing the target sample's class label.

| Neighborhood Metric | Correlation with Score |
|---|---|
| Number of same-label neighbors | $-0.2971$ |
| Number of different-label neighbors | $0.2630$ |
| Mean distance to same-label neighbors | $-0.3128$ |
| Mean distance to different-label neighbors | $0.2863$ |
| Min. distance to same-label neighbors | $-0.2303$ |
| Min. distance to a different-label neighbor | $0.2572$ |

The analysis reveals a clear pattern: samples with low importance scores typically reside in dense, class-homogeneous regions (negative correlation with same-label neighbor count). In contrast, high-importance samples are characterized by proximity to different classes (positive correlation between different-label neighbor count and distance), identifying them as informative or "hard" examples near decision boundaries.

These findings provide a strong empirical foundation for our extrapolation framework. They justify the use of distance-based interpolation and, critically, motivate the inclusion of class labels as features for the GNN model to capture these highly predictive local statistics.

## C.6 PRUNING OUTPERFORMS INTERPOLATION

To validate that our extrapolation methods are meaningfully effective and not simply benefiting from random chance or linear score mixing, we perform additional comparisons against three distinct baselines: (1) a *Score Sampling baseline*, (2) a *Naive Theoretical Interpolation*, and (3) a *Multi-Layer Perceptron (MLP) regression*.

**Score Sampling:** We implemented this baseline by assigning random scores sampled uniformly within the score range of the subset. The results Table 17 show that our extrapolations consistently outperform this baseline, confirming that extrapolation recovers meaningful structure.

Table 17: Extrapolation vs. random score sampling on ImageNet

| Method | 10% | 20% | 50% |
|---|---|---|---|
| GT DU | 0.6347 | 0.6341 | 0.5908 |
| Random | 0.6275 | 0.6211 | 0.5856 |
| 20% GT DU + Random Sampling | 0.6294 | 0.6227 | 0.5848 |
| GNNDU [Ours] | **0.6341** | **0.6288** | **0.5889** |

**Naive Theoretical Interpolation:** In a broader sense, our extrapolation could be viewed as an interpolation; for example, a subset of 100% would represent the full method with full cost, while a subset of 0% would be random with almost negligible costs. Increasing the subset size increases approximation quality and pruning performance, but also runtime. A perfect approximation would lead to an identical sample ordering.

To validate the effectiveness of extrapolation, we compare the extrapolated performance with a naive and theoretical linear interpolation between the methods and random performance. Nonetheless, such an interpolation is practically not possible and serves as a theoretical baseline only.

$$\text{Score}_{Naive} = \text{Size}_{Subset} \cdot \text{Score}_{Method} + (1 - \text{Size}_{Subset}) \cdot \text{Score}_{Random}$$

Table 18 shows that our extrapolated scores outperform the interpolated scores and are much closer to the performance of the original scoring method.

Table 18: We compare our extrapolation approach with a simple and only theoretical interpolation approach for TDDS on Places365.

| Method | 10% | 20% | 50% | 80% | 90% | 95% |
|---|---|---|---|---|---|---|
| TDDS (GT) | 0.4321 | 0.4309 | 0.4297 | 0.4119 | 0.3834 | 0.3461 |
| Random | 0.4309 | 0.4290 | 0.4214 | 0.3892 | 0.3606 | 0.3289 |
| Naive Interpolation | 0.4311 | 0.4294 | 0.4231 | 0.3937 | 0.3652 | 0.3323 |
| GNN TDDS [Ours] | **0.4319** | **0.4307** | **0.4245** | **0.3962** | **0.3683** | **0.3396** |

**Multi Layer Perceptron:** In addition, we compare our extrapolation methods with a simple two-layer Multilayer Perceptron (MLP) of the hidden dimension 256 and 64 based score regression in Table 19. While the MLP can map similar embeddings to similar scores and provides better correlation than the non-parametric KNN approach, which only interpolates neighborhood scores, it requires model training. Our GNN, on the other hand, combines neighborhood and embedding information, resulting in the highest correlation.

Table 19: Comparison of Pearson correlation on CIFAR-10 using a 20% subset.

| | MLP | KNN [Ours] | GNN [Ours] |
|---|---|---|---|
| **Correlation** | 0.3209 | 0.2765 | 0.4080 |

## C.7 Relation to Dynamic Pruning and Speedup Approaches

The main advantage our score extrapolation provides is reducing the computational effort needed to produce scores for the entire dataset. As mentioned in Section 2, other approaches address the time-to-accuracy problem. These approaches typically modify the training curriculum, such as RS2, while others also discard samples during training.

**RS2:** Repeated Random Sampling (RS2) (Okanovic et al., 2024) samples a different random subset for training each epoch. In this way, the total number of iterations per epoch and the total number of epochs per sample are reduced. Since no samples are permanently excluded or scores are calculated, it is not, in fact, a pruning method.

In Table 20, we compare RS2 against our extrapolated scores and the original (GT) scores of DU and TDDS. It can be seen that RS2 performs well in both the initial phase (10%) and at the highest pruning rate (95%), where pruning methods typically suffer from performance drops. RS2 uses all samples and only reduces the number of iterations per sample. These observations are logical, as for small pruning rates, the improvement is negligible, and for high pruning rates, the dataset's diversity is reduced, leading to performance drops. We see RS2 as an orthogonal approach to pruning, as it addresses a different direction and can also be applied to our initial training.

Table 20: Performance comparison of RS2 with the original DU and TDDS scores and our extrapolated version. The best scores among the speedup methods are bold.

| Method | 10% | 20% | 50% | 80% | 90% | 95% |
|---|---|---|---|---|---|---|
| Random | 0.7378 | 0.7339 | 0.7305 | 0.7113 | 0.6927 | 0.6738 |
| GT DU | 0.7436 | 0.7441 | 0.7544 | 0.7561 | 0.7464 | 0.7142 |
| GT TDDS | 0.7429 | 0.7438 | 0.7491 | 0.7534 | 0.7481 | 0.7251 |
| RS2 | **0.7426** | 0.7410 | 0.7387 | 0.7267 | 0.7184 | **0.7125** |
| GNN DU | 0.7416 | 0.7422 | **0.7438** | **0.7415** | **0.7287** | 0.6962 |
| GNN TDDS | **0.7426** | **0.7431** | 0.7429 | 0.7411 | 0.7250 | 0.7116 |

**InfoBatch:** InfoBatch (Qin et al., 2024) is an approach that combines altering the training, specifically the loss function, and pruning. It applies random pruning to remove less informative samples based on the loss distribution. To mimic them in the training procedure, it rescales the gradients of the remaining samples. By doing so, InfoBatch aims to maintain the original gradients.

While for RS2 we can scale the number of iterations for the samples to match the pruning rate for evaluation, a fair comparison for InfoBatch is hardly possible. For static pruning, the evaluation focuses on the reduced set to maintain similar training conditions. However, by additionally scaling the loss function, InfoBatch gains an advantage over the other methods, which are tested under similar conditions. This alteration to the training process increases the complexity for hyperparameter configuration, often necessitating a change in the optimizer or learning rate to ensure stable training under the loss scaling of InfoBatch. Of the two tested standard configurations with the Adam and SGD optimizers, only the Adam showed stable training. We compared InfoBatch with our KNN extrapolation on synthetic CIFAR-100, achieving a comparable accuracy of 0.69. Our method only requires **114 mins**, compared to InfoBatch's **342.617 mins**.

Similar to RS2, we consider InfoBatch's underlying work to be orthogonal to our score extrapolation method. Our concept focuses not on novel pruning, but on efficiently propagating scores in

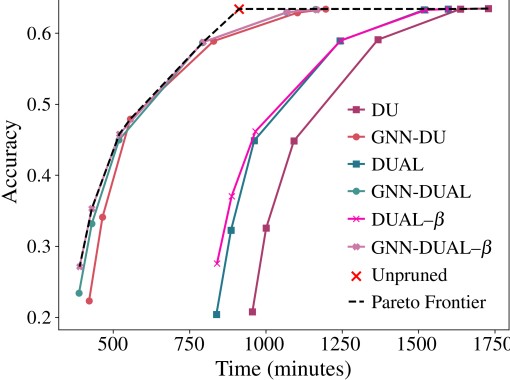

Figure 9: Comparison of DU, DUAL, and their GNN extrapolations for ImageNet on a time vs. accuracy Pareto plot. It can be seen that our extrapolation approach can benefit from both the cost reduction of DUAL and the Beta sampling advantages of DUAL. Additionally, our extrapolation retains the pruning effectiveness of the score itself.

the latent space for static score propagation. Therefore, our extrapolation approach can be combined with InfoBatch (as well as RS2) to achieve further performance gains.

**DUAL:** Difficulty and Uncertainty-Aware Lightweight (DUAL) (Cho et al., 2025), builds upon DU and optimizes the dynamic uncertainty, such that the same quality of importance scores can be reached with fewer epochs of training. In addition, DUAL applies a Beta distribution sampling strategy to increase data selection effectiveness.

In Figure 9, we compare DU with DUAL, with and without Beta sampling, with our extrapolation approach. It can be seen that our approach can effectively extrapolate the DUAL score, maintaining its effectiveness and also profiting from its speedup. In addition, we modified the Beta sampling to work with the reduced subset, aiming to boost performance on the extrapolated scores in a similar manner.

**Score Extrapolation:** Contrary to previous approaches, which either alter the training curriculum, drop samples during training, or reduce the computational cost of scoring methods, our extrapolation approach addresses computational cost reduction in a completely different direction. Our framework provides a general approach, independent of the training process, a specific task, or a scoring method. We showed that different approaches can be combined with score extrapolation. Additionally, our score extrapolation also only necessitates labels for the initially labeled subset, providing a significant advantage over other speedup methods.

