# OpenReview forum: "Effective Data Pruning through Score Extrapolation"
_ICLR.cc/2026/Conference — Submitted to ICLR 2026_

### Official Review · Reviewer_WNwP · 2025-10-27

**Soundness:** 3
**Presentation:** 2
**Contribution:** 3
**Rating:** 4
**Confidence:** 4

**Summary:**

This paper introduces a novel score extrapolation framework that eliminates the need for full training on the original dataset before pruning. The framework trains on a subset of the original data and extrapolates scores using k-nearest neighbors (KNN) and graph neural networks (GNN) based on the trained subset's scores. The method is validated across various datasets and training paradigms based on two different scoring methods. Their approach achieves Pareto-optimal time-accuracy trade-offs and significantly reduces computational effort.

**Strengths:**

1. The paper's main contribution addresses a key challenge in existing data pruning methods: ironically, these methods take longer to prune and train on a pruned dataset than to train fully on the original dataset. This paper mitigates this issue by calculating scores only on a subset of data points and extrapolating scores for the remaining data, rather than computing scores for every data point. This methodology arises from a strong motivation and solves the problem in a reasonable way.
2. The authors tested their score extrapolation methods on various datasets (Places365, ImageNet, Synthetic CIFAR-100, and CIFAR-10). They experimented across multiple settings—including supervised, unsupervised, and adversarial training—to investigate the applicability of score extrapolation methods. They also conducted various ablation studies, such as examining the correlation with original scores depending on initial sample size.
3. Importantly, the authors include failure cases in Appendix C.2 to highlight the current limitations of their framework, demonstrating transparency and a careful evaluation of their method.

**Weaknesses:**

1. Section 3 lacks the clarity necessary for comprehension and reproducibility. The main issues are as follows:
    1. What is $D$ in line 166 refer to? The notation should be clearly defined. Moreover, the symbol $D$ overlaps with the distance notation $D(\cdot, \cdot)$ used in line 210, which may cause confusion.
    2. In line 178, the authors defined the embedding function $\phi: \mathcal{X} \to \mathbb{R}^d$ that maps an input $x$ to embedding $z = \phi(x)$.
        - First, $x \in \mathcal{X}$ should be explicitly stated.
        - Second, in lines 200~212 (*the Extrapolation with KNN paragraph*), the model is defined as $\mathcal{F}_s: \mathbb{R}^d \to \mathbb{R}^{d'}$. Here, $\mathbb{R}^d$ is used as the input dimension and $\mathbb{R}^{d'}$ as the output dimension, which differs from the earlier definition.
        - Moreover, $\mathcal{F}_s$ is described as an embedding function (outputting embeddings) rather than a neural network that outputs logits, which contradicts the definition in line 178. The authors should consistently use $\phi_s$ to denote the embedding function for clarity and consistency.
    3. In Equation (2), $y$ should be explicitly defined, including its dimensionality.
    4. In lines 171 and 175, the authors define $S_s$ and $S_r$ as vectors. However, in Equation (6), the same notation $S$ is used both as a function ($S_{knn}$), and as a scalar ($S_{\pi_i(x)}$). This inconsistency becomes more pronounced in lines 230-237, where $S_s$ and $S$ are again treated as functions. If $S$ is intended to represent a “score”, its form should be clearly and consistently defined to avoid confusion for readers.
    5. The authors should explicitly state that $\pi_i(x) \in [m]$, indicating that the index of the $i$-th nearest neighbor of $x$ is selected from within $\mathbb{D}_s$.
    6. In line 221, $d(\cdot, \cdot)$ is used as a distance metric, which is inconsistent with the notation $D(\cdot, \cdot)$ defined in line 210-211.
    7. In Equation (7), the authors use $\sum_{x_i \in \mathbb{D}\_s}$, but the term $\mathcal{F}\_\mathcal{G}(\mathcal{A}, \mathcal{V};\theta)\_i$ explicitly depends on the index $i$. Therefore, I recommend replacing $\sum\_{x\_i \in \mathbb{D}\_s}$ with $\sum\_{i \in [m]}$, since the authors have already defined $\mathbb{D}\_s = \lbrace x_i \rbrace_{i \in [m]}$.

    Overall, the Section 3 should be rewritten for clarity, consistency, and reproducibility. The notations should be clearly defined and used consistently, and any ambiguous or overlapping definitions should be eliminated.

2. Comparison with recent SOTA baselines could further strengthen the proposed method:
    1. For example, the DUAL method (with its $\beta$-sampling scheme) explicitly targets reducing the score-computation time by jointly considering data uncertainty and difficulty [1].  Since your paper focuses on effective data pruning, this baseline appears directly relevant. It would be valuable either to compare your proposed method with DUAL, or to incorporate DUAL’s score as a base for your method, potentially improving overall pruning efficiency when combined with its sampling strategy.
    2. Also, methods such as Coverage-centric Coreset Selection (CCS) [2] and $\mathbb{D}^2$ [3] involve more expensive score-computation but achieve very competitive performance. These could serve as strong base scores or benchmarks for your method, especially if you aim to show improvement in computation vs. performance trade-off.

    To adopt the sampling methods from [1] and [2], one can first compute subset scores, extrapolate them to estimate the unevaluated samples, and then apply sampling based on these extrapolated scores.

3. Are there specific reasons for using the *Synthetic CIFAR-100* dataset instead of the standard CIFAR-100? Since the abstract (line 21) mentions CIFAR-100, readers may be confused when encountering Synthetic CIFAR-100 in the main text. As a reader unfamiliar with this dataset, I believe the paper should clearly explain:
    1. What this dataset is, including how it relates to the original CIFAR-100; and
    2. Why the authors chose to use this dataset instead of the standard CIFAR-100.

    My understanding is that the authors used the Synthetic CIFAR-100 dataset because the score extrapolation process requires computing scores on a subset first, and the absolute size of the dataset plays a critical role in this step. Nevertheless, I believe readers will still be interested in the model’s performance on the original CIFAR-100, especially since results on the original CIFAR-10 are already provided. Therefore, I recommend including the original CIFAR-100 results (at least in the appendix or as a stated limitation).

4. Why are there no data pruning results in the unsupervised setting? Currently, only Pearson and Spearman correlation results are reported for the CIFAR-10 dataset in Table 3. Since the scores have already been computed, it should not be difficult to perform dataset pruning based on these scores and then train the pruned subset in a supervised manner. Given that the paragraph is titled *“Unsupervised Data Pruning”,* readers (including myself) would naturally be interested in seeing the performance of a pruned subset whose scores were calculated in an unsupervised manner. I therefore recommend that the authors include the performance results of such a pruned subset.
5. As stated in Appendix C.2, there are certain datasets for which the score extrapolation method fails to accurately capture the original score distribution. Moreover, as shown in Figure 4 and Table 2, the correlations with the original scores are not particularly high. This issue is also reflected in the pruned dataset performance presented in Figure 2, where the pruning results are not significantly better than those of other pruning methods. Taken together, these observations suggest that the proposed approach behaves more like an interpolation between random pruning and full score computation, rather than a method that resolves the trade-off between computation and performance. In this regard, it would be helpful to include the results of random pruning in Figure 3 for comparison. Furthermore, I recommend that the authors to include the results of EL2N [4] and DUAL [1], which are both time-efficient data pruning methods.

---

**References:**

[1] Lightweight Dataset Pruning without Full Training via Example Difficulty and Prediction Uncertainty, ICML 2025.

[2] Coverage-centric Coreset Selection for High Pruning Rates, ICLR 2023.

[3] D2 Pruning: Message Passing for Balancing Diversity and Difficulty in Data Pruning, ICLR 2024.

[4] Deep Learning on a Data Diet: Finding Important Examples Early in Training, NeurIPS 2021.

**Questions:**

1. Why do the SSP and Zcoreset methods, which are training-free, take so much time to create a subset in Figure 3?
2. What does each point in Figure 3 represent? How were these points determined? If each point corresponds to a different pruning ratio, this should be clarified. Similarly, what does each point in Figure 4 represent? It seems likely proportional to the subset size used for score calculation, but this needs explanation.
3. When should KNN be used versus GNN? Which method is appropriate under which circumstances, and why? (Also, clarification is needed for lines 412–413. What does "This might be ~" mean exactly?)
4. In the *Adversarial Training* paragraph, as a reader unfamiliar with adversarial training, it is difficult to understand the metrics used in Table 4 and Table 5. The authors should clarify the definitions of “Robustness” and “Clean Accuracy.”
5. What if we train an independent model that maps embeddings to subset scores? Specifically, up to Step 3 in Figure 1, the process remains the same, but for the extrapolation step, we train a small neural network (e.g., an MLP) on the subset where the scores have been computed, and then evaluate it on the remaining data to predict their scores. How would such an approach perform?
6. As far as I understand, the proposed method requires labeling only the initial subset when pruning the dataset, even in the supervised setting. Is that correct? Since only the embeddings of the remaining data are used when extrapolating the scores, if the pruning can indeed be performed without access to all labels, I think the authors should highlight this point. For example: “Our method requires labels only for the initial subset, thereby reducing labeling costs and enabling application to large-scale web datasets.”
7. There appears to be overlapping information among Table 1, Figure 2, and Figure 3. The authors might consider merging or simplifying these to avoid redundancy and improve readability.
8. Is there a specific reason for reporting *relative accuracy* in Table 1? It might be more intuitive to present the *actual accuracy* values while expressing *time efficiency* in relative terms, as this would make the comparison easier to interpret.

---
**Minor Corrections:**

1. Line 149: data sets → datasets
2. Figure 2(c) and Figure 6(c): Synthic → Synthetic
---
If the authors adequately address the issues raised in the Weaknesses and Questions sections, I would consider increasing my overall score.

---

> ### Author Response · Authors · 2025-11-21
> **Weaknesses**
>
> ## Thank you
> We sincerely thank the reviewer for the helpful and detailed feedback, which contributed to improving clarity, completeness, and presentation. We address all points below and incorporate them into the updated manuscript.
>
> ## Weaknesses
>
> **W1 Clarity & Reproducibility:**
> > We acknowledge the detailed feedback on this section.
> > **Revision:** We improved the clarity of the notations, removed overlaps, and ambiguities. In addition, we improved formulations to support clarity and reproducibility (Code is available for reproducibility).
>
> **W2 Recent SOTA baselines:**
> **a) DUAL**
> >Thank you for this suggestion, indeed DUAL [1] is very interesting as it 1) improves the efficiency by requiring fewer training epochs and 2) provides a post-processing for the scores.  DUAL [1] also outperforms EL2N [4] in time efficiency and pruning performance.
> >**Revision: We added DUAL [1] as a baseline**, and show that its reduced training time and β-sampling integrate well with our extrapolation approach (App C.7.). We will provide DUAL for all Datasets in the Camera Ready version.
>
> **b)CCS, & hybrid/ geometric methods**
> >We agree these are strong baselines. However, they leverage a costly **geometric distance calculation that scales polynomially with dataset size**, which becomes the bottleneck on large datasets. Extrapolation reduces initial training and importance-score costs, but cannot reduce the inherent geometric-distance costs, thereby mitigating time advantages.
> **Revision:** We clarified this in the manuscript (Sec. 4).
>
> **W3. Synthetic vs. original CIFAR-100:**
> >Good point, we shared only a few details besides the citations.
> We added a clear explanation of the dataset and our motivation. Synthetic CIFAR-100 enables **large-scale (million-image) pruning evaluation**, which matches our target regime. Nevertheless, we agree that the original dataset is valuable.
> **Revision:** We add a explanation & descrption in App. A.1 and will include CIFAR-100 results in the Camera Ready version.
>
>
> **W4. Missing unsupervised pruning results:**
> > Importance score pruning has not been applied has not been applied yet to unsupervised classification yet. Our extrapolated scores showed high correlation and good score approximation.
> > However, since the baseline unsupervised scoring methods we tested did not yield effective pruning results,
> we focused this section on the correlation and approximation quality of the extrapolation, contrasting the supervised pruning.
> >**Revision:** We clarified and added more details in Sec. 4 and B.3.
>
>
> **W5. Interpretation and random pruning baseline:**
> > Yes, some datasets would require a) more complex models or b) an increased subset to reflect their distribution accurately. We show this more detailed in our updated App. C.2.
>
> >Yet, the main goal of the paper is to propose and validate the concept of extrapolation. Finding the best extrapolator remains for future work. Our experiments show that extrapolation can work in diverse settings and is, in all cases, more efficient than the baselines.
> > In a broader sense, our extrapolation could be seen as an interpolation, e.g., a subset of 100% would be the full method with full cost, while a subset of 0% be random. Increasing the subset size increases approximation and pruning performance, but also runtime. A perfect approximation would lead to the original sample ordering. However, 1) an interpolation is practically not possible, 2) our method achieves a better time performance tradeoff and pruning performance than a theoretical interpolation (e.g., 0.2*method score + 0.8 random score or sampling the remaining scores).
> As the table below shows, the GNN method consistently and significantly outperforms the Naive Interpolation and Random baselines, confirming that it recovers meaningful structural information and behaves more like an approximation. By this, we strongly differentiate our approach from simple interpolation or random methods.
>
> | Method | 10% | 20% | 50% | 80% | 90% | 95% |
> | :--- | :---: | :---: | :---: | :---: | :---: | :---: |
> | TDDS (GT) | 0.4321 | 0.4309 | 0.4297 | 0.4119 | 0.3834 | 0.3461 |
> | Random  | 0.4309 | 0.4290 | 0.4214 | 0.3892 | 0.3606 | 0.3289 |
> | Naive Interpolation | 0.4311 | 0.4294 | 0.4231 | 0.3937 | 0.3652 | 0.3323 |
> | GNN-TDDS  | 0.4319 | 0.4307 | 0.4245 | 0.3962 | 0.3683 | 0.3396 |
>
> >**Revision**: a) We extended the limiation in C.2, highlight the impact of different models and subset sizes, b) we include a discussion about the tradeoff and our results beyond different interpolation schemes in C6 and c) added DUAL [1] further baseline with a discussion about time effectiveness in C.7. We will update Fig. 2 & Fig.3 once all scenarios are completed.

---

> ### Author Response · Authors · 2025-11-21
> **Questions**
>
> ## Questions
>
> **Q1. SSP & ZCoreSet:**
> >They rely on geometric distance computations, which scale polynomially with dataset size, and are more expensive than training for huge-scale datasets.
>
> **Q2. Points in Fig. 3 & 4**
> >Fig. 3: different pruning ratios (commonly used values aligned with Fig.2).
> >Fig. 4: different subset sizes.
> >**Revision:** Captions and descriptions have been updated.
>
> **Q3. KNN vs. GNN**
> >KNN: fastest, preferred for efficiency.
> >GNN: higher correlation and accuracy; better for complex manifolds.
> >**Revision:** We added a brief recommendation in the discussion section after the limitations.
>
> **Q4. Adversarial metrics**
> >Thank you for pointing this out!
> > Clean Accuracy: Accuracy of the model on unperturbed test inputs (standard performance without adversarial noise).
> > Robustness: Accuracy of the model on adversarially perturbed test inputs generated by the chosen attack method. How well the model maintains correct predictions under adversarial perturbations.
> >**Revision:** We added the explanations in the manuscript.
>
> **Q5. MLP extrapolation**
> >An MLP ignores neighborhood structure. A GNN can find the optimum between self and neighboring embeddings; if neighboring is ignored, it behaves like an MLP.
> >**Revision:** A comparison is included in the App C.6..
>
> **Q6. Label requirements**
> >Correct: labels are only needed for the initial subset for kNN and the GNN variant w/o labels.
> >**Revision:** We now highlight this as an advantage.
>
> **Q7. Overlap between tables/figures**
> >We will reduce redundancy where possible while retaining necessary information, once we have the additional baselines for all experiments ready.
>
> **Q8. Relative accuracy:**
> >We used the relative accuracy because pruning quality is upper-bounded by the original scores serving as ground truth for extrapolation. Relative accuracy directly reflects this relationship.
>
> ## Minor
> >Thanks! All suggestions have been applied.

---

### Official Review · Reviewer_VYC6 · 2025-10-29

**Soundness:** 3
**Presentation:** 3
**Contribution:** 2
**Rating:** 4
**Confidence:** 3

**Summary:**

Traditional pruning methods require training computing importance scores on all the samples of the dataset. Computing these scores often requires training a model for multiple epochs on the samples, which can become costly. This work proposes to compute the scores (and train an initial model) on only a small fraction of the samples and extrapolate to the rest of the dataset. To do so, they propose two different technics, one based on K-Nearest Neighbours and the other training a Graph Neural Network, both using the geometry induced by the embedding space.

**Strengths:**

In general I found this work interesting, it tackles a less explored direction in data pruning and can indeed bring valuable computational gains.
- The notion of extrapolating importance scores from a small subset is simple but original (to my knowledge), and it provides a new angle on making pruning efficient.
- The KNN and GNN approaches effectively demonstrate that extrapolation can cut computation time with little performance loss and form a good proof of concept.
- The work evaluates multiple datasets, and training settings, which supports the generality of the approach.

**Weaknesses:**

Overall, the paper presents an interesting idea, though I think it could be strengthened with some further development and analysis.
- I did not find the theoretical justification very convincing. It seems that the main point is about the smooth interpolation of the samples influence that the authors use as a justification for their extrapolated scores (eq 6). But in the context of influence the extrapolated point is itself a convex interpolation of the reference points, and the weights are the associated factors, whereas in eq 6 we are assigning scores using an arbitrary formula depending on the distance to the reference points. Also the authors could introduce what is influence and why it would be relevant here.
- I think the paper could benefit from a comparison with other simple alternatives to reduce computations (see questions).
- Dynamic data pruning methods are not considered in this work, though they are very popular and somehow connected to this since they face the problem of updating the scores and can do it on only a fraction of the samples every time to avoid prohibitive costs.

**Questions:**

Multiple "simple" alternatives could be compared to your proposed approach to strengthen the contribution
- Instead of training completely on the whole dataset, how would the proposed approach compare to training only a few epochs and use this intermediate model to score, then keep training the same model with only the selected fraction, which is often what is done in practice.
- Could one use directly your model $\mathcal{F}_s$ trained on the selected subset and score the unseen points without using extrapolation ?
- In table 1, how would you compare to using the "ground truth scores" for the selected subset and random for the unseen, without needing extrapolation ?

- In line 152-153, the authors write that the trained model $\mathcal{F}_s$ depends on the selected extrapolation method, but from lines 167-172 it seems to be independent of the extrapolation method, could you clarify ?

---

> ### Author Response · Authors · 2025-11-21
> **Weaknesses**
>
> ## Thank you
>
> We sincerely thank the reviewer for the thoughtful and encouraging comments for improving our manuscript. We are glad that you found the idea interesting and the experimental evaluation comprehensive. We address your concerns and describe the revisions below.
>
> ## Weaknesses
>
>  **W1. Theoretical justification and influence functions**
> >Thank you for raising this conceptual point. You are right that the interpolation property of influence **does not imply** that influence can be computed using a simple distance-based formula.
> The classical influence-function result states that, under smoothness assumptions, the influence of a training point on the loss at an **interpolated evaluation point** is itself approximately a convex interpolation of the influences at the endpoints. This means:
> >* Influence varies **smoothly** in the embedding space.
> >* Nearby points tend to have **similar influences**.
> >* But influence is *not* a radial function of Euclidean distance.
> Thus, our extrapolation in Eq. (6) is **not intended as a direct influence estimator**. Instead, we treat influence theory as providing a *qualitative* structural property—namely, that importance is locally smooth. Distance is, therefore, a convenient and practical proxy for enforcing smoothness when propagating importance scores from a small scored subset to the rest of the dataset.
>
> >**Revisions:**
> >* We explicitly clarified this distinction and reworked the text to distinguish clearly between “theoretical motivation” (smoothness) and “practical extrapolation” in Sec. 3.
> >* We added a background on influence function Sec. 3, explaining what influence functions capture, why their smoothness motivates our approach, and how this aligns with our goal of identifying which samples are most important for data pruning.
>
> **W2. Simple alternatives:**
> >Thank you for the recommendation. Some approaches, like DUAL [A] (see questions), reduce the training time for the scoring model.
> We provided experiments for all of your questions, underlining that a) reduction effects can be combined with our extrapolation and b) our approach outperforms "simpler methods".
> >Curriculum-based approaches (e.g., RS2 (Okanovic et al., 2024)) reduce training cost, but **do not prune the dataset**. We see these as orthogonal to (classic) pruning and focus on approaches that actually reduce the dataset instead of modifying the training strategy.
> **Revision:** We added experiment with DUAL (See Q1) & other approaches suggested in your Questions.
>
>
> **W3. Dynamic data pruning**
> >We appreciate your excellent suggestion. We agree that dynamic data pruning methods represent a relevant class of techniques.
> However, we note that these methods either a) update the training set during training or b) switch samples in a rolling batch dynamically w/o reduction (see W2). To drop samples during training, dynamic pruning approaches alter the training curricula, optimizer, and loss, which makes a fair comparison more difficult.
> >Since our primary goal is to propose and evaluate the novel concept of extrapolation as a general approach, independent of a specific importance score, task, or training scheme, and not propose a novel pruning method, we focused on the evaluation of this concept for different (static) importance score methods, including adversarial robustness.
> >Thus, our method can be seen as orthogonal to pure data pruning and used together with dynamic data pruning techniques to potentially further enhance performance, serving as an additive layer.
> >Nevertheless, the concept of dropping samples dynamically can serve as a related baseline, so we added a discussion and experimental study in this direction.
> Our discussion shows that the mentioned variations in the training schema not only make a fair comparison difficult but also introduce extra complexity for InfoBatch [C], as the training process itself changes. Nevertheless, we compared InfoBatch with our extrapolation on Synthetic CIFAR-100 under similar hyperparameter settings, which highlights the time efficiency of our method. At a comparable accuracy of 0.69, our KNN extrapolation requires 114 minutes while InfoBatch [C] takes 343 minutes.
> >**Revision:** We added a discussion of dynamic pruning, its relation to extrapolation, and provided additional experiments in App C.7. We will extend the results in the updated version.

---

> ### Author Response · Authors · 2025-11-21
> **Questions**
>
> ## Questions
>
> **Q1. Reducing training epochs**
> >This approach and an analysis of epochs to score quality is done by DUAL [A], which requires fewer epochs for scoring the samples. Yet, these effects also hold for our reduced initial subset. We report acc and time [s] below
>
> | Method | 10% | 20% | 50% | 80% | 90% | 95% |
> | :--- |  :---: | :---: | :---: | :---: | :---: | :---: |
> | **DU** | 0.6347 / 1729 | 0.6341 / 1638 | 0.5908 / 1367 | 0.4484 / 1092 | 0.3255 / 1001 | 0.2078 / 956 |
> | **GNN-DU** | 0.6341 / 1196 | 0.6288 / 1103 | 0.5889 / 829 | 0.479 / 556 | 0.341 / 465 | 0.2232 / 421 |
> | **DUAL**  | 0.6339 / 1597 | 0.6329 / 1519 | 0.5891 / 1243 | 0.4488 / 962 | 0.3227 / 886 | 0.2040 / 838 |
> | **GNN-DUAL**  | 0.6335 / 1163 | 0.6292 / 1068 | 0.5869 / 792 | 0.4496 / 519 | 0.3318 / 430 | 0.2342 / 388 |
> | **DUAL-$\beta$**  | 0.6341 / 1599 | 0.6326 / 1523 | 0.5898 / 1246 | 0.4617 / 966 | 0.3705 / 888 | 0.2758 / 840 |
> | **GNN-DUAL-$\beta$**  | 0.6331 / 1167 | 0.6296 / 1069 | 0.5875 / 795 | 0.4579 / 520 | 0.3530 / 430 | 0.2712 / 390 |
>
> >**Revision**: We added experiments with DUAL, which show that a) our training can similarly be shortened (additive effect), and DUAL and their Beta sampling can be extrapolated with high quality. Experiment for initial revision in App C.7, and will be included in the main paper after we obtain results for all settings.
>
> **Q2. Scoring unseen points w/o extrapolation**
> >Extrapolation is the step of assigning a score for the unseen points, either with our KNN or GNN (extrapolation is a simple forward pass) approach, so w/o extrapolation, they have no scores.
> >Sampling is another possibility (see Q3).
> Also, an MLP which can learn to map similar embeddings to similar scores, yet it is less effective than a GNN (correlation 0.3209 vs 0.4080).
> **Revision:** We added this to App C.6.
>
>
> **Q3. Ground-truth (GT) scores + random sampling**
> >Thanks for this great sanity check idea.
> We implemented this baseline by assigning random scores sampled uniformly within the score range of the subset. Results show that our extrapolations consistently outperform this baseline,
> confirming that extrapolation recovers meaningful structure.
>
> |Method|10%|20%|50%|
> |---|---|---| ---|
> |Random|0.6275|0.6211|0.5856|
> |GT DU|0.6347|0.6341|0.5908|
> |:---|:----:|:----:|---:|
> |20% GT DU + Random Sampling|0.6294|0.6227|0.5848|
> |**GNN [Ours]**|**0.6341**|**0.6288**|**0.5889**|
>
> >**Revision:** We added a detailed table & discussion about simple scoring methods to App C.6.
>
> **Q4. Clarifying lines**
> >Thanks for catching this misleading formulation. L.167-172 are correct, 152–153 should express a freely selectable scoring method.
> >**Revision:** We updated this formulation.
>
> [A] Lightweight Dataset Pruning without Full Training via Example Difficulty and Prediction Uncertainty, Cho et. al., ICML 2025
> [B] Instance - Dependent Early Stopping, Yuan et al., ICLR 2025
> [C] InfoBatch: Lossless Training Speed Up by Unbiased Dynamic Data Pruning, Qin et. al., ICLR 2024

---

### Official Review · Reviewer_mc1X · 2025-10-31

**Soundness:** 3
**Presentation:** 2
**Contribution:** 3
**Rating:** 6
**Confidence:** 3

**Summary:**

This paper introduces a novel and efficient approach to data pruning. The method first estimates the importance of each training example by analyzing only a small subset of the data. It then extrapolates these importance scores to the entire dataset using one of two proposed strategies: one based on k-nearest neighbors (KNN) and the other on a graph neural network (GNN). Both strategies leverage embeddings from a model trained exclusively on the small subset. The authors evaluate their approach on four large-scale image datasets, covering supervised learning, unsupervised learning, and adversarial training scenarios. There are comprehensive experiments compare the method against state-of-the-art pruning techniques and simple baselines. Results demonstrate that the proposed approach significantly reduces computational cost while achieving accuracy that matches or closely approaches that of existing methods.

**Strengths:**

- The paper tackles a practically significant and under-addressed problem: how to make computationally expensive data pruning methods tractable for large-scale training by requiring only a small subset for direct score computation.
- The proposed score extrapolation framework is methodologically interesting and is instantiated with both a simple, transparent KNN approach and a more expressive, message-passing-based GNN, allowing for a clear analysis of trade-offs.
- Empirical validation is thorough across multiple datasets—CIFAR-10, CIFAR-100, Places-365, and ImageNet—and considers a range of pruning rates, tasks (supervised, unsupervised, adversarial), and multiple pruning methods (DU and TDDS).
- The efficiency benefit is systematically explored, with computing time and accuracy jointly visualized. As demonstrated in **Table 1**, the proposed KNN and GNN approaches recover a substantial portion of the original pruning benefit at a fraction of the computational cost, achieving notable speedups.

**Weaknesses:**

1. **Limited theoretical justification and over-reliance on local linearity assumptions:**
   The primary mathematical support for extrapolation is drawn from influence function and local linearity arguments (Section 3). Yet, there is insufficient theoretical development or empirical diagnosis regarding the validity of these assumptions for highly nonlinear, high-dimensional representation spaces found in deep learning. As such, generalizability of the approach to broader architectures/tasks remains open.
2. **Score oversmoothing and limited modeling of complex distributions:** A main weakness shows up in **Figure 5** (Appendix C.2). Both the KNN and GNN methods tend to oversmooth the extrapolated importance scores. They miss the multiple peaks that exist in the original score distribution. This makes the approach less effective when data importance isn’t uniform—for example, when there are clearly two different groups of samples with different levels of importance. It can also lead to consistent ranking mistakes, especially for outliers or samples that are hard to classify. The paper notes these issues, but it doesn’t fix them with new experiments or method changes.
3. **Modest accuracy improvement and moderate correlation, especially at small subset sizes:** The extrapolation quality drops when the subset is too small or when a lot of data is pruned. This holds true both for how well the scores match the true importance and for the final model accuracy.  In many cases, even the best extrapolation method doesn’t fully match the performance of the original pruning approach. The results also change noticeably depending on the size of the subset used.  The method does offer real speedups. However, it doesn’t always give a better balance between accuracy and resource use. In settings with very tight resource limits, this could be a serious limitation.
4. **Mathematical construction and empirical transparency:**
   Some mathematical steps and notation—in particular, the loss formulations for the GNN extrapolation and details of embedding construction—are not explained in enough depth (e.g., Section 3, Equation for GNN loss). The explanation of exactly how node features combine (what happens with missing labels in unsupervised, for instance), and the optimization details, are relegated to appendices and could benefit from greater clarity in the main paper.
5. **Potential failure cases insufficiently investigated:**
   The brief exploration of failure modes (e.g., rank differences related to background/outliers in **Figure 5**) is insightful but limited. More systematic error analysis—including quantifying what data characteristics lead to the highest extrapolation mis-rankings or when the approach could harm downstream performance—is warranted.

**Questions:**

1. Could the authors elaborate on the limitations of the local linearity assumption in embedding space? Specifically, how does this assumption break down for highly heterogeneous data distributions, and are there diagnostics or empirical controls to quantify this risk in practice?
2. In GNN extrapolation, how are class labels handled for semi-supervised or unsupervised settings, especially when no reliable pseudo-labels are available? Would the approach degrade under severe class imbalance or label noise?
4. What strategies do the authors propose to mitigate oversmoothing and better capture multimodal or long-tailed importance score distributions in future work?
5. How does the extrapolation performance vary with increasing/heterogeneous dataset size (e.g., simulated “billion-sample” settings), or for modalities outside vision (e.g., text, multimodal tasks)?

---

> ### Author Response · Authors · 2025-11-21
> **Weaknesses**
>
> ## Thank you
>
> We sincerely thank the reviewer for the constructive feedback and for highlighting the novelty and practical value of our approach. Below, we address each weakness and provide improvements.
>
> ## Weaknesses
>
> **W1. Theoretical justification and local linearity**
> >Thank you for raising this concern. Our theoretical arguments serve as a motivation for an empirical strategy. Our intention was not to claim that influence functions provide a *direct* theoretical foundation for distance-based extrapolation.
> Influence theory only suggests that **importance varies smoothly in representation space**, not that it is a radial or linear function of distance.
> Our extrapolation methods adopt this smoothness as a **practical prior**: nearby samples tend to share similar importance.
> While the KNN variant is tightly tied to this assumption, the GNN allows deviations from strict local linearity by learning more flexible propagation patterns.
> >**Additional evidence in C.5:**
> >* **Correlation with local neighborhood:**: We observed consistent relationships between sample importance and properties such as label homogeneity and intra-class distances. DU scores on ImageNet show consistent relationships with neighborhood structure (dense homogeneous regions → low importance, boundary regions → high importance).
> >* **FID analysis:** We evaluated how well different sampling methods preserve the distribution of the original dataset using FID. Subsets of our approach show strong distributional alignment with the full data.
>
> >Together with the strong correlation and the final performance reported in the main paper, these results provide empirical support that our assumptions hold well in practice across diverse datasets and architectures.
>
> >**Revision:** We clarified this distinction in Sec 3, added a short influence background, and elaborated the analysis in Appendix C2 and C5.
>
> **W2. Oversmoothing**
> >We agree that oversmoothing can occur, especially with small subsets or lightweight extrapolation models (e.g. our small GNN).
> We start with an i.i.d. sample subset which captures the underlying distribution (Robert & Casella, 2004). By increasing the (i) subset size or the (ii) GNN complexity, the multimodal approximation ability improves. Architectures like residual GNNs [A] mitigate oversmoothing on graphs. Both (i) and (ii) increase costs. Also, further hyperparameter tuning would likely strengthen the extrapolation results. Still, the aim of this paper is to establish the validity of score extrapolation rather than to optimize every component. Our method already surpasses the baselines, which supports the core claim without additional tuning.
>
> >**Revision:** We added the oversmoothing to the limitation and enriched the analysis in C.2 with additional statistics and subset sizes.
>
> **W3. Accuracy and correlation:**
> >You are right that very small subsets reduce extrapolation fidelity. This reflects an inherent trade-off: smaller subsets capture less structure but offer more speed. At high pruning ratios, pruning scores drop sharply, which naturally affects the extrapolated scores as well. Even so, we observe a clear and stable relation between extrapolated and ground truth performance. Our extrapolation outperforms random pruning whenever the underlying methods do, as shown in Fig. 2 and Fig. 6. While absolute accuracy can decline slightly, the time–accuracy balance remains consistently strong. Across all settings, our methods stay **Pareto-optimal** (Fig. 3).
> >Could you please elaborate under which conditions you see accuracy/resource balance at risk?
>
> >**Revision:** We added a comparison with naive interpolations, underlining the quality of our extrapolation.
>
> **W4. Mathematical details**
> >We appreciate the opportunity to improve our manuscript.
>
> >**Revision:** We added 1) an explicit description of embedding construction, 2) clarified GNN loss formulation, and 3) moved essential details to Sec. 3.
>
> **W5. Failure cases analysis**
> >We extended our failure case analysis by additional statistics relating extrapolation to model confidence. In addition, we provided results for a further subset size and discussed model correlations in this context. It can be seen that difficult/atypical samples also suffer from a bad extrapolation and the mitigation through appropriate subset sizes.
>
> >**Revision:** We extended the setting of Fig.5 as described for Q2 and provided additional statistics, including confidence relations.

---

> ### Author Response · Authors · 2025-11-21
> **Questions**
>
> ## Questions
>
> **Q1. Limitations of the local-smoothness**
> > We refer to W1 for clarification on the assumptions.
> Highly heterogeneous datasets increase complexity, which can be captured by increasing our initial subset size (Robert & Casella, 2004) or more complex GNNs (See W2/Q3 with revisions). For diagnostics, confidences, training correlation or distances between the samples can be used to get a deep understand how the extrapolation would perform.
>
> **Q2. Labels in GNN extrapolation**
> >When labels are unreliable (or unavailable), the labels are omitted for the GNN. We performed a study on the impact of adding labels, which increased the correlation of DU on ImageNet from 0.3925 to 0.4193 (see App B.4.). Using labels improves the correlation as the model can capture label-dependent properties, but label noise would degrade performance. However, given this limited impact, it is not strictly required for the extrapolation framework to be effective.
> >**Revision:** We discussed the impact of labels in App B.4.
>
> **Q3. Mitigating in future work**
> >Graph-based formulations enable the integration of oversmoothing-resilient architectures (e.g., residual GNNs [A]).
> >**Revision:** We discuss this idea as future work.
>
> **Q4. Behavior on large or multimodal sets**
> >To simulate large datasets, we used the million-scale synthetic CIFAR-100, providing a comparable sample-to-class ratio.
> We found that extrapolation has difficulty in improving beyond the full trained model performance, but shows high speedups and pruning performance (Fig. 2c).
> However, we require a) a score calculation on all data as a baseline for benchmarking, and b) scoring methods used to extrapolate that do not consider heterogeneous sets or other modalities, which limits the choice of suitable datasets.
>
> [A] Residual Connections and Normalization Can Provably Prevent Oversmoothing in GNNs, Scholkemper et al.,ICLR 2025
>
> ## Final Remarks
>
> Thank you again for the insightful feedback. We believe the added clarifications, expanded analyses, and improved methodological descriptions significantly strengthen the paper.

---

### Official Review · Reviewer_uwUo · 2025-10-31

**Soundness:** 2
**Presentation:** 2
**Contribution:** 2
**Rating:** 2
**Confidence:** 4

**Summary:**

This paper addresses the computational efficiency problem in data pruning methods by proposing a novel score extrapolation paradigm. The core idea is to compute importance scores on a small subset (10-20%) of the training data and then extrapolate these scores to the remaining unseen samples using KNN or GNN-based methods. The authors demonstrate that this approach achieves up to 4.9× speedup while maintaining competitive downstream task performance across supervised, unsupervised, and adversarial training settings on datasets including ImageNet, Places365, CIFAR-10, and synthetic CIFAR-100.

**Strengths:**

1. Using the Extrapolation method to refine the sample ranking is rather new.

**Weaknesses:**

1. The critical weakness is the overlooked training cost.  The method requires: 1) First training: Train on random 10-20% subset to compute initial scores. 2) Embedding + Extrapolation: Extract features and extrapolate scores for remaining 80-90%. 3) Second training: Train final model on the extrapolated-pruned subset.

1.1 This means training happens TWICE on similar-sized subsets, plus embedding the full dataset.

1.2 Did the authors account for the cost of BOTH training phases? The paper claims "4.9× speedup" but it's unclear if both training runs are included in the time measurement.

1.3 What is the cost of embedding 100% of the data? This requires forward passes through the entire dataset, which is not negligible.

1.4 What is the actual extrapolation cost? KNN search or GNN inference on large-scale datasets has computational overhead.

1.5
The paper should clearly demonstrate that:
 ```
Cost(Training 20% + Embedding 100% + Extrapolation + Training 20%)
<
Cost(Training 100% once + Standard pruning)
```

2. The code provided by the authors cannot be opened.

3. On line 752, the authors mentioned "For the unsupervised setting, we employ DINOv2 (Oquab et al., 2023) as a foundation model to obtain fixed embeddings for all samples. ". Why not use this as a baseline and see how much the supervised method can improve?

**Questions:**

See weaknesses

---

> ### Author Response · Authors · 2025-11-21
>
> # Q1. Training and Computation Cost
> Thank you for highlighting this important aspect. Efficiency is central to our contribution, and we appreciate the opportunity to clarify.
> **All reported speedups (Fig. 3, Tab. 1) include every step** of our pipeline, comprising:
>
> Standard pruning:
> 1. Train on 100% of data $T_{\text{train100}}$
> 2. Score 100% $T_{\text{score100}}$
> 3. Train on pruned subset for evaluation $T_{\text{prune}}$
>
> Our extrapolation:
> 1. Train on subset (e.g., 20%) $T_{\text{train20}}$
> 2. Score only subset $T_{\text{score20}}$
> 3. Embed all data $T_{\text{embed100}}$
> 4. Extrapolate scores $T_{\text{extra}}$
> 5. Train on pruned subset for evaluation $T_{\text{prune}}$
>
> The final comparison is therefore:
>
> $ \text{Cost}_{\text{extrapolation}}=$
>
>  $T_{\text{train20}}+T_{\text{score20}}+T_{\text{embed100}}+T_{\text{extra}}+T_{\text{prune}}$
>
> vs.
>
> $ \text{Cost}_{\text{standard}}=$
>
> $T_{\text{train100}}+T_{\text{score100}} +T_{\text{prune}}$
>
> The mentioned **second training** ($T_{\text{prune}}$) is part of the evaluation and is common to standard pruning methods and equally expensive.
>
> **Q1.1 Twice the training**:
> >The second training is **on the pruned set for evaluation**, which is a **common evaluation step also required for standard pruning approaches**.
> For set selection, all methods require only one training, and only the **first** training differs (e.g., ours: 20% vs. original: 100% of the data).
> Importantly, if the pruning ratio and the subset size are both 20% (as in your example), both training steps involve **different** sets.
> **Revision**: We updated the misleading text, which caused a lack of clarity in Sec 4.
>
> **Q1.2 Reported Times**:
> >Yes, the reported times include both steps and measure the time for all steps shown above.
> Fig. 3 shows a Pareto plot (accuracy vs. time) of total costs (**every step**), highlighting the significant time reduction of our extrapolation in all scenarios, across different pruning ratios, methods, and datasets.
> The 4.9x improvement is based on results in Tab 1: our KNN with a 10% subset size, extrapolated TDDS on Places365, and a pruning rate of 95%.
> **Revision**: We highlighted this explicitly.
>
> **Q1.3 embedding cost**:
> >Embedding requires a single forward pass over the dataset, which is **substantially cheaper** than full training. Assuming the costs of forward and backward passes to be equal, creating embeddings is cheaper than one epoch if the subset is less or equal than 50%. However, the training usually consists of 50-200 epochs, making the embedding calculation negligible.
> Empirically, this is reflected in Fig. 3, where extrapolation is consistently Pareto-optimal.
> **Revision:** See revision Q1.x
>
> **Q1.4 Extrapolation cost**:
> >* **KNN** is (O(n\cdot d)) and lightweight relative to model training.
> >* **GNN** model size is significantly smaller than the task model (ResNet 50 ~23.5m vs. GNN ~1.4m parameters). Also, the training on the graph based on the embedding and neighbor sampling is more effective and requires fewer iterations than the task model.
> **Revision:** We added a comparison of model size and KNN complexity for KNN in the paper in an extended cost discussion in App A.6.
>
> **Q1.5 Verifying the inequality**:
> >We agree that this is fundamental. As discussed in Q1.3 and Q1.4, embeddings and extrapolation are more effective than training many epochs of the full dataset.
> Fig. 3 explicitly plots **full pipeline runtime vs. accuracy**, demonstrating that our approach is Pareto-optimal across datasets, while standard pruning is not, such that the inequality is fulfilled.
> **Revision:** See revision Q1.x
>
> >**Revision for Q1.x:**
> We improve the description of the Time Optimality section and the paragraph describing Tab.1, providing a deeper explanation of all time components and the evaluative nature of the second training. In addition, we add a detailed discussion of all steps and their costs in the added sec. A.6., including a runtime consideration of kNN and GNN based on model size.
>
> # Q2. Code availability:
> >Thanks for pointing this out. The link's expiration was accidental. Please accept our apologies.
> **Revision:** We have restored the link and will replace it with a proper GitHub link in the final version.
>
> # Q3. Why not use DINOv2 as a supervised baseline?:
> > Thanks for this question! The DINOv2 features are part of the **unsupervised** classification method Turtle (Gadetsky et al., 2024) described in line 401.
> For **supervised pruning**, we follow established in-domain setups (e.g., ResNet backbones) for comparability to prior works (DU, TDDS).
> We also ran preliminary tests with SSCD [A] and DinoV2 (ViT-g/14) as embedding models, but using pretrained embedding models decreased final training performance in most cases (yet, differences were minor). More importantly, they introduce a dependency on pretrained models, which we aim to avoid.
>
> [A] A Self-Supervised Descriptor for Image Copy Detection, Pizzi et. al., 2022, arXiv:2202.10261

---

### Author Response · Authors · 2025-11-21
**Summary and thanks to all reviewers**

We sincerely thank all reviewers for their thorough and constructive feedback. We are pleased to see strong alignment regarding the **key contributions** of our work. Reviewers consistently highlighted:

* the **novelty** and **originality** of refining sample ranking through extrapolation [uwUo, VYC6], addressing key limitations of existing approaches [WNwP];
* the **practical significance** of making expensive pruning methods tractable by computing scores on only a small subset [mc1X];
* the **simplicity, generality, and expressiveness** of the proposed KNN and GNN extrapolation strategies [mc1X, VYC6];
* the ability to achieve **substantial computational savings** while preserving the benefits of full-score pruning [mc1X, VYC6];
* the **breadth of the empirical evaluation** across datasets, training regimes, and pruning algorithms [mc1X, VYC6, WNwP];
* and the **transparent reporting of limitations and failure cases** [mc1X, WNwP].

We deeply appreciate the reviewers’ thoughtful feedback. We want to emphasize that we have **addressed all raised concerns**, including:

* **Clarification of cost and time estimation** (Sec. 3), including a full cost breakdown of embedding creation & extrapolation vs full training (App. A.7.) [uwUo W1] and elaboration that our methods are indeed **always Pareto-optimal** (Fig.3) [mc1X W3];
* **Clearification on experimental design choices** [uwUo W2, WNwP W4], **dataset choices** (Sec. 4 & App. B.3.) [WNwP W3] and presentation [WNwP Q1-Q4, Q6-Q8, VYC6 Q4];
* **Elaboration of the theoretical justification**, a clearer presentation in Sec. 3, resolution of notation issues (Sec. 3), and enriched the empirical evidence (App C.2.) [mc1X W1,W4;VYC6 W1; WNwP W1]
* **Comparison and discussion of additional pruning baselines, dynamic pruning and time-saving approaches**, including DUAL [A], RS2 [B] and InfoBatch [C] (Sec 4, App C.7.) [WNwP W2; VYC6 W3];
* **Comparisons with simpler extrapolation, interpolation, sampling strategies, and MLP**, as well as expanded discussion on applicability to broader settings (App. C.6.) [VYC6 W2,Q1–Q3; WNwP Q5];
* **Additional experiments and discussion on failure modes, limitations, and oversmoothing** (Sec 4, App. C.2.) [WNwP W5; mc1X W2,W5];
* Resolution of remaining questions and minor issues, including the expired repository link [uwUo W2].

All points are addressed thoroughly in the individual reviewer responses, supported by new explanations, expanded discussions, and additional experiments where needed. While not all new experiments could be conducted across every setting, due to the **broad scope** of our evaluation (itself noted as a strength), the results demonstrate the benefits of our approach across all requested additions. Per the official guidance, we submit the revised manuscript within the first week, with all changes marked in blue.

Overall, these clarifications and additions meaningfully strengthen the paper and further substantiate its contributions. We thank the reviewers again for their constructive engagement, which has significantly improved the quality of our work, and we hope that the improvements, combined with our numerous strengths, including the highlighted **novelty of our concept and practical significance**, make a compelling case for acceptance.

[A] Lightweight Dataset Pruning without Full Training via Example Difficulty and Prediction Uncertainty, Cho et. al., ICML 2025
[B] Repeated Random Sampling for Minimizing the Time-to-Accuracy of Learning, Okanovic et. al, ICLR 2024
[C] InfoBatch: Lossless Training Speed Up by Unbiased Dynamic Data Pruning, Qin et. al., ICLR 2024

---

### Meta-Review · Area_Chair_cVYU · 2025-12-24

**Summary:**

This paper proposes a novel score extrapolation framework to data pruning tasks. The core idea is to compute the importance score on a small subset of training data, and then extrapolate the scores to the remaining samples using KNN or GNN-based methods. Extensive experiments have shown the effectiveness of the framework, providing new directions for data pruning tasks.

The reviewers raise several core concerns including (1) the lack of a verifiable theoretical mechanism behind the score extrapolation assumption and the reliance on strong empirical intuitions, (2) the unclear conceptual novelty of the method, which appears closer to an efficiency-oriented approximation technique rather than a new data pruning principle, and (3) the reliability and generality of extrapolated scores when used for high-risk pruning decisions, especially under limited experimental scale and settings.

The authors’ rebuttal attempts to address the reviewers’ concerns and some of them have been answered clearly. However, several outstanding issue remain: (1) the core extrapolation assumption is still supported mainly by empirical observations rather than a testable or mechanism-level justification, (2) the extrapolated scores may not be sufficiently reliable to support irreversible pruning decisions, given the observed modest performance and moderate correlations, and (3) the claimed scalability and general applicability are not fully validated due to the restricted model, dataset, and task coverage.

In summary, while the paper presents a simple and well-motivated framework for accelerating data pruning through score extrapolation, several key issues remain insufficiently addressed despite the authors’ rebuttal. These unresolved concerns limit the strength of the paper’s theoretical grounding, practical reliability, and claimed scope of applicability. As a result, I think that the current version of the paper is below the acceptance bar of ICLR.

**Reviewer Concerns:**

In the rebuttal, the authors have addressed some of the reviewer concerns, including:

- Training cost and efficiency: The authors clarified the previously overlooked training cost analysis in their method. They provided a clearer breakdown of the computation involved in score estimation and extrapolation. Additional empirical results were added to report actual running time. These clarifications make the efficiency claims more concrete and easier to assess.

- Comprehensive baseline comparisons: The authors expanded the experimental comparisons with more baseline methods. They included simple alternatives and random pruning as reference points. Dynamic data pruning methods were incorporated, making the evaluation more complete.

- In-depth analysis and ablations: The rebuttal added several analyses to better explain the experimental results. The authors conducted a failure case analysis to show when the method performs poorly. They justified the use of Synthetic CIFAR-100 instead of standard CIFAR-100. Missing unsupervised pruning results and discussions on modest accuracy gains and moderate correlations were added.

- Presentation and reproducibility: The authors have refined the paper based on reviewer feedback. Several formulations, justifications, and citations were clarified or corrected. The overall presentation is more structured and easier to follow. The release of source code further improves reproducibility and transparency.

However, there are still some concerns that are failed to be fully addressed:

- Empirical core assumption: The proposed score extrapolation relies on an empirical and intuition-driven assumption about score transferability. This assumption is not formulated as a testable or mechanism-level hypothesis. The experimental results only provide indirect support under specific settings. There is no clear explanation of why the extrapolation should hold in general, which weakens the theoretical grounding of the method.

- Strong assumptions behind the Pareto-optimal claim: The paper claims Pareto-optimal efficiency between computation cost and pruning quality. However, this claim relies on strong and implicit assumptions about training dynamics and score stability. The paper lacks strict comparisons with alternative efficiency–accuracy trade-offs.

- Closer to an engineering acceleration technique: The method primarily reuses existing pruning scores and focuses on reducing their computation cost, which makes the contribution closer to an engineering optimization than a conceptual advance in pruning methods.

- Generalization remains unverified: The experiments are conducted on a narrow set of models and datasets. Score extrapolation behavior may strongly depend on specific training dynamics and architectures. And there is little evidence that the method generalizes across different tasks or model scales.

**Reviewer Scores:**

- Reviewer uwUo: 2->4

Justification: While the authors addressed training cost and presentation issues, the request for a DINOv2-based baseline comparison was not directly resolved, leaving the reviewer’s core experimental concern largely unchanged.

- Reviewer mc1X : 6->6

Justification: The rebuttal added several analyses and ablations, but the limited theoretical justification and the modest performance gains still constrain the perceived strength of the contribution.

- Reviewer VYC6 : 4->4

Justification: Although additional baselines and clarifications were provided, the theoretical explanation remains unconvincing and the method still appears closer to an engineering technique than a new pruning principle.

- Reviewer WNwP : 4->4

Justification: The authors responded to most experimental and presentation-related comments, but the moderate correlations and interpolation-like behavior limit upward revision of the score.

---

### Decision · Program_Chairs · 2026-01-26

Reject